# Real-World Image Super-Resolution as Multi-Task Learning

**Wenlong Zhang[1,2], Xiaohui Li[2,3], Guangyuan Shi[1], Xiangyu Chen[2,4,5]**
**Xiaoyun Zhang[3], Yu Qiao[2,5], Xiao-Ming Wu[1][†] Chao Dong[2,5][†]**
[1]The HongKong Polytechnic University [2]Shanghai AI Laboratory
[3]Shanghai Jiao Tong University [4]University of Macau
[5]Shenzhen Institute of Advanced Technology, CAS
`wenlong.zhang@connect.polyu.hk, xiao-ming.wu@polyu.edu.hk, chao.dong@siat.ac.cn`

## Abstract

In this paper, we take a new look at real-world image super-resolution (real-SR) from a multi-task learning perspective. We demonstrate that the conventional formulation of real-SR can be viewed as solving multiple distinct degradation tasks using a single shared model. This poses a challenge known as task competition or task conflict in multi-task learning, where certain tasks dominate the learning process, resulting in poor performance on other tasks. This problem is exacerbated in the case of real-SR, due to the involvement of numerous degradation tasks. To address the issue of task competition in real-SR, we propose a task grouping approach. Our approach efficiently identifies the degradation tasks where a real-SR model falls short and groups these unsatisfactory tasks into multiple task groups. We then utilize the task groups to fine-tune the real-SR model in a simple way, which effectively mitigates task competition and facilitates knowledge transfer. Extensive experiments demonstrate our method achieves significantly enhanced performance across a wide range of degradation scenarios. The source code is available at `https://github.com/XPixelGroup/TGSR`.

## 1 Introduction

**Real-world image super-resolution** (real-SR) aims to enhance the resolution and quality of low-resolution images captured in real-world scenarios. Real-SR algorithms enable improved image quality and better visual understanding, making them valuable in a wide range of applications [43, 46, 45, 14, 17]. Unlike non-blind SR [9, 56, 55] and classical blind SR [13, 18, 38] that assumes a simple degradation process, real-SR deals with complex and unknown degradations present in real-world imaging conditions, such as noise, blur, compression artifacts, and sensor limitations. The diversity of real-world degradations and unknown degradation parameters make it challenging to reverse the specific degradation effects and accurately recover high-resolution details.

**Prior studies** tackle real-SR by designing various degradation models to simulate the degradation process in real-world scenarios. For instance, they generate synthetic paired training data with a shuffled degradation model (BSRGAN [50]), a high-order degradation model (RealESRGAN [39]), or a three-level degradation model (DASR [25]). Existing methods commonly train a single real-SR network with training data generated by a sophisticated degradation model, aiming to cover as many degradation cases as possible during the training process. Under this framework, recent works have focused on enhancing various aspects of the real-SR network, including improving the backbone network architecture [26], optimizing inference efficiency [25], enhancing generalization capabilities [23], and improving modulation ability [31].

---

[†]Corresponding author

37th Conference on Neural Information Processing Systems (NeurIPS 2023).

**This study takes a new look at real-SR from a multi-task learning perspective.** We show that the conventional formulation of real-SR essentially corresponds to a multi-task learning problem, which involves solving a large number of different degradation tasks simultaneously with a single shared model. Consequently, real-SR faces a well-known challenge in multi-task learning, namely *task competition or task conflict* [32]. It refers to the situation that tasks compete for model capacity, potentially resulting in certain tasks dominating the learning process and adversely affecting the performance of other tasks. This problem is severely aggravated in the case of real-SR, where a large number of degradation tasks (up to thousands) are typically involved. In our pilot experiment (Fig. 1), we have observed that a real-SR network trained with the multi-task objective fails to yield satisfactory results for a significant portion of degradation tasks.

**We propose a task grouping approach** (Fig. 2) to alleviate the negative impact of task competition for real-SR. Task grouping [48, 37, 35, 10] is an effective technique in multi-task learning that helps mitigate negative transfer by training similar tasks together. This approach typically involves learning the relationship or relevance between pairs of tasks through validation or fine-tuning, which becomes impractical in the context of real-SR where there is a large number of tasks to consider. Hence, we resort to design indirect measures to assess task affinity. Specifically, we introduce a performance indicator based on gradient updates to efficiently identify the degradation tasks that a real-SR model falls short of. Then, we propose an algorithm to group these unsatisfactory tasks into multiple groups based on a performance improvement score. Finally, we propose TGSR, a simple yet effective real-SR method, which leverages the identified task groups to fine-tune the pre-trained real-SR network. Our key contributions are summarized as follows:

- We take a new look at real-SR from a multi-task learning perspective and highlight the task competition problem.

- We propose a task grouping approach to effectively address the task competition problem for real-SR and develop a task grouping based real-SR method (TGSR).

- We conduct extensive experiments to validate the effectiveness of TGSR, and the results demonstrate its superior performance compared to state-of-the-art real-SR methods.

## 2 Related Work

**Image super-resolution.** Since Dong *et al.* [9] first introduced Convolutional Neural Networks (CNNs) to the SR problem, a series of learning-based SR methods have made great progress, including deep networks [21], dense connections [56], channel attention of [55], residual-in-residual dense blocks [41], and transformer structure [26]. To reconstruct realistic textures, Generative Adversarial Networks (GANs) [24, 41, 53, 42] have been employed to generate visually pleasing results. However, these methods adopt a bicubic down-sampling degradation that is insufficient for real-world images.

**Real-world super-resolution.** To address the SR problem in real-world scenarios, classical blind SR methods primarily use Gaussian blur and noise to model the distribution of real-world images. Significant progress has been made through a variety of approaches, including the use of a single SR network with multiple degradations [51], kernel estimation [13, 18, 44, 2], and representation learning [38]. Additionally, BSRGAN [50] proposes a large degradation model that incorporates multiple degradations using a shuffled strategy, while RealESRGAN [39] employs a high-order strategy to construct a large degradation model. DASR [25] adopts a three-level degradation distribution (i.e., two one-order and one high-order degradation models) to simulate the distribution of real-world images. These works demonstrate the potential of large degradation models for real-world applications. In addition, recent works also improve real-SR in multiple dimensions, such as network backbone [26], efficiency [25], generalization [23, 54, 52] and modulation ability [31].

**Multi-task learning**. Multi-task learning methods can be roughly divided into three categories: task balancing, task grouping, and architecture design. Task balancing [16, 20, 47, 27, 34, 15, 8, 7, 19] methods address task/gradient conflicts by re-weighting the loss or manipulating the update gradient. Task grouping [48, 37, 35, 10] methods mainly focus on identifying which tasks should be learned together. Zamir et al. [48] provide a task taxonomy that captures the notion of task transferability. TAG [10] determines task groups by computing affinity scores that capture the effect between tasks. Architecture design methods can mainly be divided into hard parameter sharing methods [22, 28, 3] and soft parameter sharing methods [30, 33, 12, 11]. Hard parameter sharing methods require

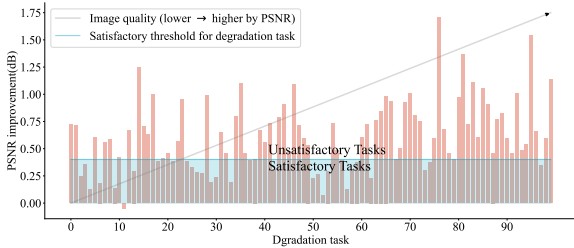
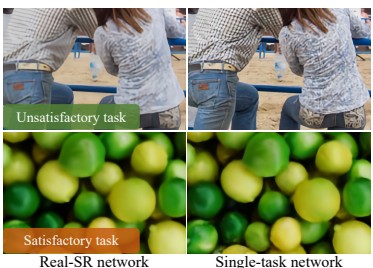

(a) PSNR distance between the real-SR and single-task SR networks across 100 randomly selected degradation tasks.

(b) Visual comparison between satisfactory and unsatisfactory tasks.

Figure 1: **Illustration of task competition.** The jointly-trained multi-task real-SR network falls short in producing satisfactory results for almost half of the degradation tasks, on which the fine-tuned single-task networks obtain more than 0.4dB performance gain, indicating these tasks are dominated by other tasks during the learning process.

different decoders for different tasks, while soft parameter sharing methods do cross-talk between different task networks. We focus on a task grouping strategy in our TGSR. The strategy identifies unsatisfactory degradation tasks in a large degradation space. Subsequently, we can further improve the overall performance of the real-SR network by fine-tuning it in the unsatisfactory tasks.

## 3 Real-SR as a Multi-task Learning Problem

**Problem Formulation**    Real-SR aims to restore a high-resolution (HR) image $x$ from its low-resolution (LR) counterpart $y$ that has undergone an unknown and intricate degradation process:

$$y = \mathcal{D}(x) = (f_n \circ \cdots \circ f_2 \circ f_1)(x), \tag{1}$$

where $f_i$ represents a degradation function such as Gaussian blurring with a kernel width [0.1, 2.4], adding Gaussian noise with a noise level [1, 25], a downsampling operation with a scale factor $r$, or applying the JPEG compression. Hence, $\mathcal{D}$ represents a vast continuous degradation space that encompasses an infinite number of degradations. Existing real-SR models such as BSRGAN [50], RealESRGAN [39], and DASR [25] make different assumptions in the generation of $\mathcal{D}$, aiming to simulate the highly complex degradation process in real-world scenarios.

An SR task $\tau$ can be defined as a training pair $(x, y = d(x))$ formed by sampling a degradation $d$ from the degradation space $\mathcal{D}$ and applying it on an HR image $x$ to generate an LR image $y$. Due to the infinite size of $\mathcal{D}$, it is impossible to consider every degradation case. Hence, a common approach is to sample a large number of degradations to sufficiently represent the degradation space. Given a set of high-resolution images $\mathcal{X}$, with $N$ different degradations sampled from $\mathcal{D}$ ($N \gg |\mathcal{X}|$), we can generate a set of SR tasks: $\mathcal{T} = \{\tau_i = (x_i, y_i)\}_{i=1}^N$, where $x_i \in \mathcal{X}$ and $y_i$ is obtained by applying a degradation on $x_i$. A real-SR model is commonly trained by minimizing the empirical risk:

$$\mathcal{L}_{\text{total}}(\theta) = \sum_{i=1}^N \mathcal{L}_i(\tau_i; \theta), \tag{2}$$

where $\mathcal{L}_i$ is the loss on task $\tau_i$, and $\theta$ are the trainable model parameters shared by all $N$ tasks. Therefore, real-SR is essentially a multi-task learning problem, aiming to solve $N$ different SR tasks with a single shared model. In the following, we refer to $\tau_i$ as a *degradation task*.

**Task Competition**    From the view of multi-task learning, real-SR is a challenging problem as it intends to solve a large number of different degradation tasks (e.g., $N = 10^3$) altogether with a single model, inevitably suffering from task competition that some tasks may dominate the training process leading to poor performance on other tasks [37].

To illustrate the impact of task competition, we randomly sample 100 degradation tasks with the degradation model in [39] and train a real-SR network with the multi-task objective in Eq. 2. Next, we fine-tune the real-SR network on each degradation task independently and obtain 100 fine-tuned

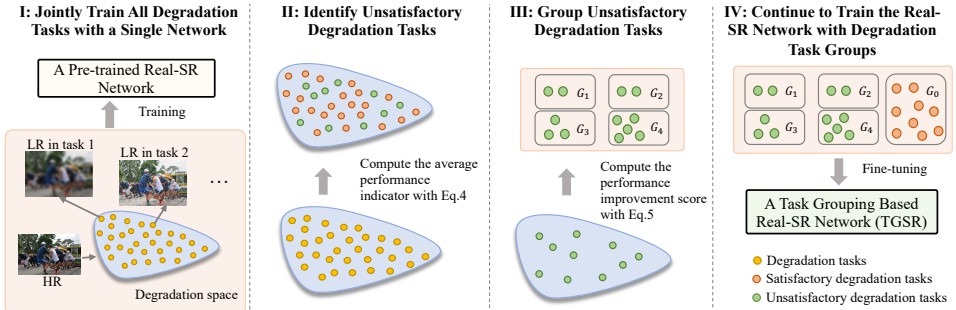

Figure 2: Overview of our proposed task grouping approach for real-SR.

models, which we call single-task networks. We then compare the performance between the real-SR network and the single-task networks on each degradation task by computing their PSNR distance. The results in Fig. 1 show that for nearly half of the degradation tasks, the PSNR distance between the real-SR network and single-task network exceeds 0.4dB, indicating that these tasks are not well solved by the real-SR network, which we refer to as *unsatisfactory tasks*. From an optimization perspective, the other tasks (those with PSNR distance less than 0.4dB) dominate the learning process and are effectively solved by the real-SR network, which we refer to as *satisfactory tasks*.

## 4 Real-SR via Task Grouping

Our analysis in Sec. 3 shows that when a real-SR network is tasked with many degradation tasks, they may compete for model capacity or interfere with each other, resulting in a significant decline in performance for certain tasks. This phenomenon, commonly referred to as *negative transfer* [37], is a well-known challenge in the field of multi-task learning. An effective approach to mitigate negative transfer is task selection or task grouping [10, 37, 48]. By finding groups of tasks that may benefit from training together, the interference among tasks can be minimized.

### 4.1 Which Tasks Should Be Learned Together for Real-SR?

The results in Fig. 1 suggest that the satisfactory tasks should be grouped together. These dominant tasks may share certain characteristics or similarities that lead to minimal task conflicts, making them more prominent in the training process. More importantly, considering that the real-SR network has effectively tackled these tasks and reached a satisfactory performance level, we may focus on the unsatisfactory tasks that present greater potential for improvement.

**Efficiently Identifying Unsatisfactory Tasks**   To identify the unsatisfactory tasks, we may use the approach described in Sec. 3 and Fig. 1, by comparing the performance between the jointly-trained multi-task real-SR network (referred to as the *pre-trained* real-SR network hereafter) and the fine-tuned single-task networks. However, the cost of adopting this approach is prohibitive due to the large number of degradation tasks (e.g., thousands or tens of thousands) involved in real-SR.

Therefore, to efficiently identify the unsatisfactory tasks, we propose a measure to assess whether a degradation task has been well solved by the real-SR network, which is done through fine-tuning the pre-trained real-SR network for a small number of iterations (e.g., 100) on the degradation task. Specifically, we assess the impact of the gradient update of task $\tau_i$ on the shared parameters $\theta$ (i.e., the pre-trained real-SR network), by comparing the loss of $\tau_i$ before and after updating $\theta$. We develop a *performance indicator* defined as:

$$z_i^t = \frac{\mathcal{L}_i(\tau_i, \theta_i^t)}{\mathcal{L}_i(\tau_i, \theta)}, \tag{3}$$

where $\theta_i^t$ are the parameters updated on task $\tau_i$ at the time step $t$. Notice that a high value of $z_i^t$ indicates the loss of task $\tau_i$ is not significantly reduced after fine-tuning, suggesting that the pre-trained real-SR network has achieved good performance on this task.

**Algorithm 1:** Degradation Task Grouping for Real-SR

---

**Input:** A set of degradation tasks $\mathcal{T} = \{\tau_1, \tau_2, ..., \tau_n\}$, the pre-trained real-SR model $\theta$, the number of groups $c$, and the threshold values $t_0, t_1, \cdots, t_c$.

**for** *any* $\tau_i \in \mathcal{T}$ **do**
   | Compute the average performance indicator $\hat{z}_i$ with Eq. 4;
**end**

// `Select unsatisfactory tasks`
Let $\hat{\mathcal{T}} = \{\tau_i | \hat{z}_i > t_0\}$ ;
// `Group unsatisfactory tasks`
**for** $i = 1, \ldots, c$ **do**
   Fine-tune the pre-trained real-SR network with all unsatisfactory tasks;
   **for** *any* $\tau_j \in \hat{\mathcal{T}}$ **do**
      | Compute the performance improvement score $s_j$ with Eq. 5;
   **end**
   Let $G_i = \{\tau_j | s_j > t_i\}$, where $t_i$ is the threshold for group $i$;
   Let $\hat{\mathcal{T}} = \hat{\mathcal{T}} \backslash G_i$;
**end**
**Output:** Degradation task groups $\mathcal{G} = \{G_1, G_2, .., G_c\}$.

---

For stability, we consider the average performance indicator during the fine-tuning process on task $\tau_i$:

$$\hat{z}_i = \frac{1}{t_e - t_s} \sum_{t=t_s}^{t_e} z_i^t, \tag{4}$$

where $t_s$ and $t_e$ represent the starting and end points of a time span, respectively. With the computed average performance indicator of each task, we can set a threshold to select the unsatisfactory tasks. The effectiveness of $\hat{z}_i$ is validated in Sec. 5.2 and Sec. 5.3.

**Grouping Unsatisfactory Tasks into Multiple Task Groups** The number of identified unsatisfactory degradation tasks can still be significant, typically in the range of hundreds in our experiments. Given the potential variance among these tasks, it would be advantageous to further divide them into multiple task groups based on their similarity to reduce negative transfer. A common approach for task grouping in multi-task learning is to learn pairwise performance indicator between tasks [48, 10, 37], which is impractical for real-SR due to the large number of degradation tasks involved.

A feasible way is to learn an indirect measure of task similarity by fine-tuning the pre-trained real-SR network on each degradation task independently and computing the PSNR distance as described in Sec. 3 and Fig. 1. However, this approach is still time-consuming given the large number of unsatisfactory tasks. Another alternative of indirect measure is the average performance indicator in Eq. 4, which, although efficient, may not be accurate enough. Hence, we make a trade-off to fine-tune the pre-trained real-SR network $\theta$ on all unsatisfactory tasks simultaneously through joint-training to obtain a new network $\hat{\theta}$. Then, we test the fine-tuned network $\hat{\theta}$ on each degradation task with an available validation set and compute a *performance improvement score (PIS)* defined as:

$$s_j = I(\mathcal{D}_j^{\text{val}}; \hat{\theta}) - I(\mathcal{D}_j^{\text{val}}; \theta), \tag{5}$$

where $\mathcal{D}_i^{\text{val}}$ represents the validation set of an unsatisfactory task $\tau_i$, $I$ is an IQA (image quality assessment) metric such as PSNR, and $s_i$ is the PIS of task $\tau_i$. We then select the tasks with PIS larger than some threshold to form a task group, which should have small conflicts as they dominate the fine-tuning process. We repeat this process to find the rest of task groups as described in Alg. 1.

### 4.2 TGSR: A Task Grouping Based Real-SR Method

With the identified degradation task groups $\mathcal{G} = \{G_1, G_2, .., G_c\}$, we adopt a straightforward method to fine-tune the pre-trained real-SR network, referred to as TGSR. We first form $\hat{\mathcal{G}} = \{G_0, \mathcal{G}\} = \{G_0, G_1, G_2, .., G_c\}$, where $G_0$ represents the entire degradation space and serves for preventing

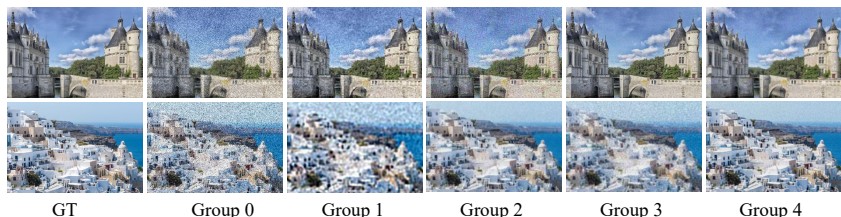

| GT | Group 0 | Group 1 | Group 2 | Group 3 | Group 4 |

Figure 3: Sample images from different degradation task groups in our DIV2K5G datasets.

catastrophic forgetting during the fine-tuning process. Then, we randomly select a group from $\hat{\mathcal{G}}$ and randomly sample a task from the chosen group for fine-tuning. This approach effectively increases the inclusion of the unsatisfactory tasks during the fine-tuning process and weights their likelihood of being chosen based on their respective group size (i.e., tasks in smaller groups have a higher probability to be selected). In essence, it is similar to the task re-weighting approach commonly used in multi-task learning.

## 5 Experiments

### 5.1 Experimental Setup

**Datasets and evaluation.** We employ DIV2K [1], Flickr2K [1] and OutdoorSceneTraining [40] datasets to implement our task grouping algorithm and train the TGSR network. For evaluation, we use DIV2K validation set to construct a DIV2K5G dataset consisting of 5 validation sets according to the divided 5 different degradation groups by the task grouping algorithm, as shown in Fig. 3. Each validation set contains 100 image pairs. In addition to the task group based test set, we employ the synthetic test set AIM2019 [29] and DIV2K_random for evaluation. The DIV2K_random is generated using the RealESRGAN degradation model on the DIV2K validation set. Furthermore, we incorporate the real-world test set, RealSR set [4], into our evaluation process. All evaluations are conducted on ×4 SR and PSNR is computed on the Y channel of YCbCr color space.

**Implementation details**. We adopt the high-order degradation model proposed by RealESRGAN [39] as the real-SR degradation model in our experiments. To compute the performance indicator, $N$ $(4 \times 10^3)$ degradation tasks are sampled from the whole degradation space. The single-task network is fine-tuned from the pre-trained RealESRNet model for 100 iterations. The performance indicator is computed based on the average of the last 10 iterations considering the instability of the training procedure. To implement the degradation grouping, the degradation tasks with the top 40% higher performance indicators ($1.6 \times 10^3$ degradation tasks) are selected as unsatisfactory tasks. Then, we fine-tune the pre-trained RealESRNet for $1 \times 10^4$ iterations based on all unsatisfactory tasks. After that, we evaluate the model on Set14 [49] for each unsatisfactory degradation task. By using the proposed performance improvement score, we divide the degradation tasks into five groups based on the thresholds of [0.8, 0.6, 0.4, 0.2]. According to this set of thresholds, we obtain four groups with the number of tasks in each group being [14, 29, 84, 200] as groups1-4. Then, we label the entire degradation space beyond the four degradation groups as Group0. To avoid forgetting the satisfactory tasks, we uniformly sample degradation tasks from each group to fine-tune the pre-trained real-SR network. Other basic training settings follow RealESRGAN.

### 5.2 Comparison with State-of-the-Art

We compare our TGSR with the state-of-the-art methods, including SRGAN [24], ESRGAN [41], RDSR [23], MM-RealSR [31] BSRGAN [50], SwinIR [26], RealESRGAN [39], DASR [25] and HAT [6, 5]. Officially released pre-trained models are used for the compared methods.

**Our TGSR improves the overall performance.** The quantitative results of different methods are presented in Tab. 1. ESRGAN is based on a single-degradation model (i.e., Bicubic setting), so it performs worst under the multi-degradation evaluation system. RDSR is an MSE-based method, so it achieves the highest PSNR but performs rather poorly (the second worst) performance on LPIPS. BSRGAN, RealSwinIR, and DASR employ a one-order degradation model. They sacrifice performance on PSNR for better perceptual quality (reflected in low LPIPS). MM-RealSR and

Table 1: Quantitative results of different methods on DIV2K5G. Group0 denotes the validation set with satisfactory degradation tasks, and Group1-4 represent the validation sets with unsatisfactory degradation tasks. The ESRGAN trained on the non-blind setting and RDSR trained on the MSE-based setting are marked in gray.

| | Group0 | | Group1 | | Group2 | | Group3 | | Group4 | |
|---|---|---|---|---|---|---|---|---|---|---|
| | PSNR | LPIPS | PSNR | LPIPS | PSNR | LPIPS | PSNR | LPIPS | PSNR | LPIPS |
| ESRGAN | 21.49 | 0.6166 | 20.29 | 0.6697 | 20.95 | 0.6530 | 22.42 | 0.5940 | 22.02 | 0.5837 |
| RDSR | 25.00 | 0.5196 | 21.18 | 0.6167 | 23.06 | 0.5586 | 25.00 | 0.4993 | 25.58 | 0.4814 |
| MM-RealSR | 23.49 | 0.4549 | 19.74 | 0.5326 | 21.76 | 0.4731 | 23.37 | 0.4205 | 23.97 | 0.3989 |
| DASR | 23.87 | 0.4683 | 19.97 | 0.5752 | 22.31 | 0.5093 | 24.00 | 0.4474 | 24.85 | 0.4225 |
| BSRGAN | 23.99 | 0.4549 | 20.07 | 0.5799 | 22.45 | 0.4961 | 24.34 | 0.4388 | 24.80 | 0.4210 |
| RealSRGAN | 23.88 | **0.4599** | 20.10 | 0.5586 | 22.12 | 0.5019 | 23.85 | 0.4499 | 24.48 | 0.4270 |
| RealSRGAN-TG | **23.95** | 0.4617 | **21.14** | **0.5323** | **23.06** | **0.4802** | **24.69** | **0.4248** | **25.05** | **0.4112** |
| RealESRGAN | 23.85 | 0.4325 | 20.10 | 0.5355 | 22.07 | 0.4701 | 24.30 | 0.4147 | 24.58 | 0.3970 |
| RealESRGAN-TG | **23.99** | **0.4286** | **21.10** | **0.5056** | **23.15** | **0.4494** | **24.62** | **0.3975** | **25.03** | **0.3851** |
| RealSwinIR | 23.35 | 0.4468 | 19.60 | 0.5624 | 21.65 | 0.4905 | 23.66 | 0.4265 | 24.16 | 0.4077 |
| RealSwinIR-TG | **23.90** | **0.4168** | **20.62** | **0.4925** | **22.70** | **0.4377** | **24.56** | **0.3815** | **24.98** | **0.3670** |
| RealHAT | 24.26 | 0.4084 | 20.64 | 0.5022 | 22.44 | 0.4468 | 24.42 | 0.3918 | 25.02 | 0.3734 |
| RealHAT-TG | **24.31** | 0.4110 | **21.41** | **0.4932** | **23.20** | **0.4395** | **24.87** | **0.3841** | **25.32** | **0.3674** |

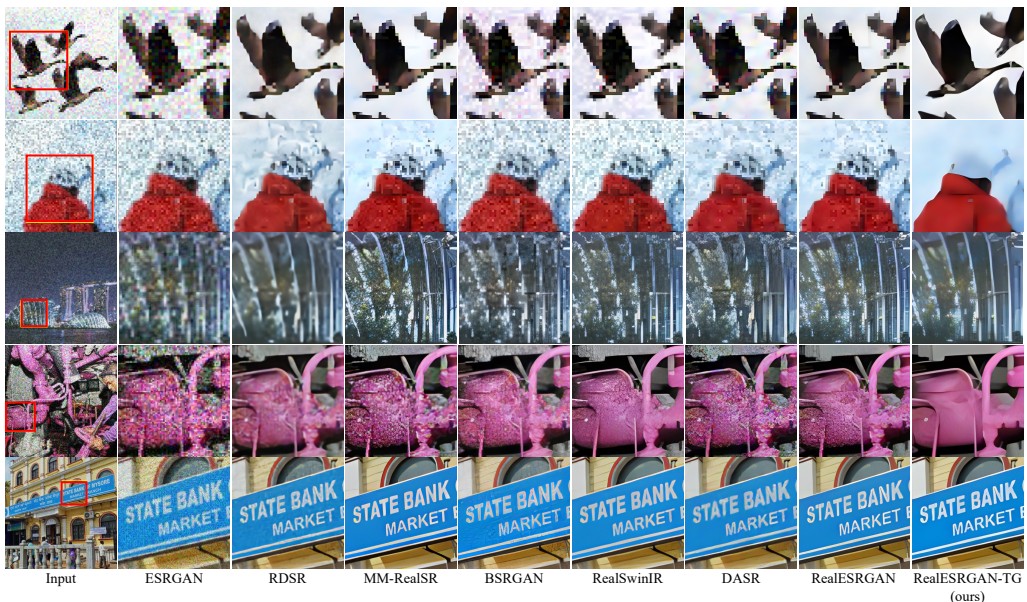

Figure 4: Qualitative results of different methods. Zoom in for details.

RealESRGAN utilize a more complex high-order degradation. As a result, they achieve better LPIPS for real-SR evaluation. However, the two approaches obtain low PSNR performance due to the great difficulty of optimizing. Notably, we can enhance the performance of the pre-trained real-SR network by fine-tuning it on the identified Task Groups (TG). This improvement is particularly significant in Groups 1-4. For instance, on RealESRGAN-TG, we can observe a maximum boost of 1 dB in PSNR and 0.03 in LPIPS.

**Our TGSR demonstrates significant superiority when applied to a large degradation space.** It is noticeable that RealESRGAN-TG outperforms RealESRGAN even in Group0. This indicates that our approach enhances real-SR performance across nearly the entire degradation space, not just the specifically addressed Groups 1-4. In addition to the identified task groups, our TGSR can also achieve performance gains on the randomized synthetic test sets DIV2K_random and AIM2019 test set as shown in Tab. 2. Additionally, our method achieves a gain on the real scene test set RealSRset. These results clearly demonstrate that our approach does not compromise the performance of certain degradation tasks to enhance the performance of others.

Table 2: Quantitative results of different methods on the real-world test set. The ESRGAN trained on the non-blind setting and RDSR trained on the MSE-based setting are marked in gray.

| | DIV2K_random | | AIM2019 | | RealSRset-Nikon | | RealSRset-Cano | |
| | PSNR | LPIPS | PSNR | LPIPS | PSNR | LPIPS | PSNR | LPIPS |
|---|---|---|---|---|---|---|---|---|
| ESRGAN | 20.63 | 0.6345 | 23.16 | 0.5500 | 27.40 | 0.4132 | 27.73 | 0.4054 |
| RDSR | 24.61 | 0.5268 | 24.44 | 0.4803 | 26.39 | 0.4053 | 26.93 | 0.3795 |
| MM-RealSR | 23.17 | 0.4394 | 23.48 | 0.3917 | 23.78 | 0.3841 | 24.42 | 0.3664 |
| DASR | 23.52 | 0.4832 | 23.76 | 0.4210 | 26.68 | 0.3972 | 27.68 | 0.3792 |
| BSRGAN | 23.76 | 0.4622 | 24.20 | 0.4000 | 26.11 | 0.3900 | 26.90 | 0.3648 |
| SwinIR | 23.13 | 0.4432 | 23.89 | 0.3870 | 26.20 | 0.3616 | 26.68 | 0.3469 |
| RealESRGAN | 23.54 | 0.4423 | 23.89 | 0.3960 | 25.62 | 0.3820 | 26.06 | 0.3629 |
| RealSRGAN | 23.58 | 0.4710 | 23.72 | 0.4247 | 24.71 | 0.4159 | 25.42 | 0.3902 |
| RealSRGAN-TG | **23.79** | **0.4705** | **23.97** | **0.4174** | **25.18** | **0.4018** | **25.93** | **0.3759** |
| RealESRGAN | 23.54 | 0.4423 | 23.89 | 0.3960 | 25.62 | 0.3820 | 26.06 | **0.3629** |
| RealESRGAN-TG | **23.84** | **0.4368** | **24.27** | **0.3899** | **26.01** | **0.3819** | **26.33** | 0.3637 |
| RealSwinIR | 23.67 | 0.4216 | 23.98 | 0.3804 | 25.70 | **0.3700** | 26.43 | **0.3506** |
| RealSwinIR-TG | **23.74** | **0.4190** | **24.10** | **0.3766** | **26.36** | 0.3751 | **27.18** | 0.3585 |
| RealHAT | 24.04 | **0.4156** | 24.19 | 0.3742 | 25.88 | **0.3532** | 26.56 | **0.3339** |
| RealHAT-TG | **24.21** | 0.4189 | **24.41** | **0.3723** | **26.12** | 0.3635 | **26.69** | 0.3451 |

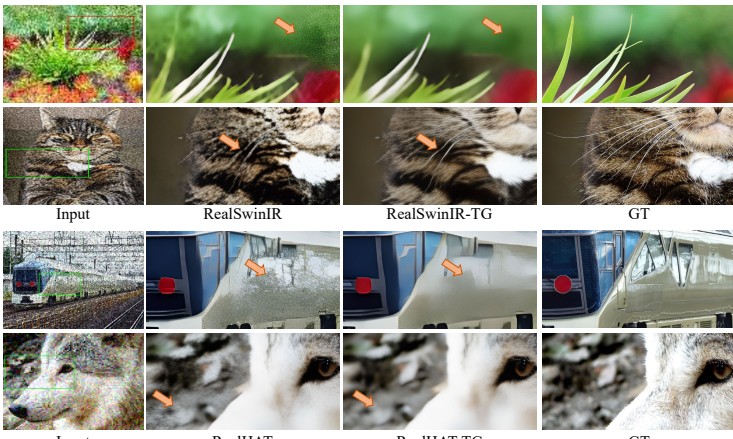

Figure 5: RealSwinIR and RealHAT may unreasonably generate unpleasant artifacts or semantic textures that shouldn't be there. However, our method does not have this problem.

**Our TGSR obtains better visual results than other methods.** The visual results of different methods are shown in Fig. 4 and Fig. 5. TGSR significantly improves existing methods by removing various degradation artifacts and noticeable unpleasant artifacts. For the first four rows of images with complex degradation, the other methods cannot remove the degradation or generate unacceptable artifacts. In contrast, our TGSR handles the degradation well and produces visually pleasant results. For the image on the last row, our method can generate more realistic results than other methods with clear textures. In the RealSwinIR results of Fig. 5, we can observe the unpleasant artifacts in the flat area of *Flower* image and the fur of *cat* image. In the RealHAT results, we also find semantic textures appearing where they shouldn't be. For example, a *tree* texture appears on the *train*, and fur-like textures are present in the background of the *Wolf* image. However, our method can remove these artifacts by fine-tuning the real-SR network on identified task groups.

## 5.3 Ablation and Analysis

**Effectiveness of the performance indicator. 1)** In Fig. 6 (a) and (b), we present the PSNR improvement achieved by fine-tuning the real-SR network on individual tasks with lower and higher performance indicators. It can be observed that tasks with higher performance indicators exhibit a significant PSNR improvement (about 0.4-1.4dB), while tasks with lower performance indicators

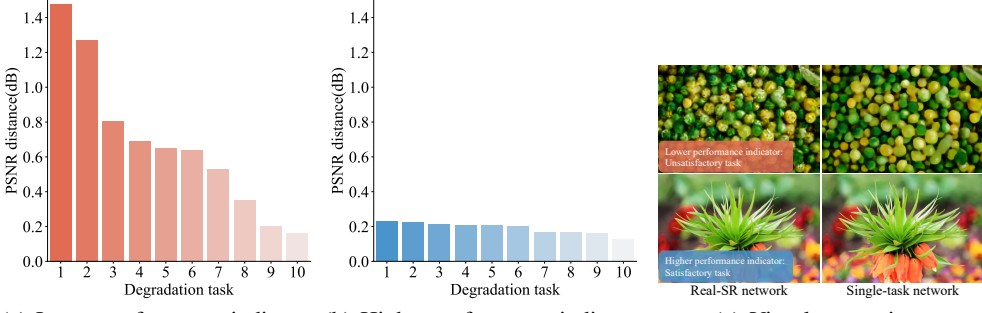

(a) Lower performance indicator    (b) Higher performance indicator    (c) Visual comparison

Figure 6: Performance comparison of real-SR networks that trained on the degradation tasks with (a) lower performance indicators and (b) higher performance indicators. The degradation tasks with a lower performance indicator can be further improved in quantitative and qualitative results.

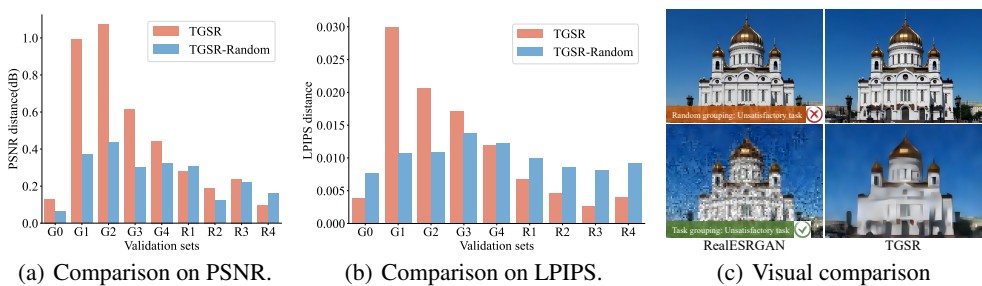

(a) Comparison on PSNR.    (b) Comparison on LPIPS.    (c) Visual comparison

Figure 7: Performance comparison of RealESRGAN and our TGSR/ TGSR with random grouping on (a) PSNR and (b) LPIPS. The results indicate random grouping achieves limited improvement compared with our proposed task grouping. (c) Visual results demonstrate that random grouping chooses a satisfactory degradation task that is not required for further training, while our task grouping method finds an unsatisfactory degradation task that needs to be further improved.

show only a small PSNR improvement (about 0.2 dB). This finding indicates that our proposed performance indicator can effectively distinguish between satisfactory and unsatisfactory tasks for a real-SR model in a large degradation space. Additionally, Fig. 6 (c) illustrates that significant visual improvements can be achieved with the unsatisfactory degradation task. **2)** Directly fine-tuning a model for each task incurs a high computational cost (empirically requiring at least 10,000 iterations). However, our performance indicator requires only 100 iterations of fine-tuning. It suggests that utilizing the performance indicator can be 100 times faster than direct fine-tuning for distinguishing the unsatisfactory and satisfactory degradation tasks. This further illustrates the superiority of the proposed performance indicator.

**Effectiveness of our task grouping algorithm.** We compare the performance of RealESRGAN-TG with our task grouping and random grouping to demonstrate the effectiveness of our method. As shown in Fig. 7 (a) and (b), random grouping can only bring very limited performance gains of about 0.2dB on PSNR and about 0.01 on LPIPS. In contrast, the groups generated by our task grouping algorithm show a significant improvement up to 1dB on PSNR and 0.03 on LPIPS. These results demonstrate the significance of a well-designed task grouping approach. Furthermore, Fig. 7 (c) shows that random grouping can not select the unsatisfactory task, while our task grouping algorithm can find it.

**Study of the performance upper bound. (1)** We randomly select 100 single tasks and then add the corresponding small range of similar degradation parameters to each task, named single-task network with range (see details in appendix). Fig. 8 shows that the existence of related tasks can improve the upper bound of some tasks (about 40%). However, there has been a performance drop for other tasks due to similar degradation parameters that may not strictly represent similar tasks. **(2)** To further show the superiority of our TGSR, we fine-tune the RealESRGAN model on each group to obtain

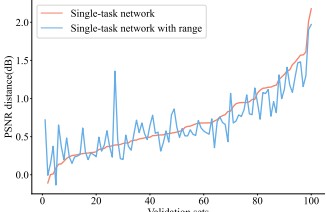

Figure 8: Performance comparison of SR network with a single task and a *range* of tasks.

Table 3: Ablation experiments on the performance upper bound. We fine-tune RealESRGAN directly on each task group to get the corresponding performance upper bound, denoted as RealESRGAN-SG.

| Metrics | Model | Group0 | Group1 | Group2 | Group3 | Group4 |
|---|---|---|---|---|---|---|
| PSNR (↑) | RealESRGAN | 23.85 | 24.58 | 24.00 | 22.07 | 20.10 |
| | RealESRGAN-SG | 23.63 | 24.58 | 24.39 | 22.67 | 20.46 |
| | RealESRGAN-TG (ours) | 23.88 | 25.08 | 24.62 | 23.11 | 21.06 |
| LPIPS (↓) | RealESRGAN | 0.4325 | 0.3970 | 0.4147 | 0.4701 | 0.5355 |
| | RealESRGAN-SG | 0.4242 | 0.3792 | 0.3929 | 0.4478 | 0.4990 |
| | RealESRGAN-TG (ours) | 0.4291 | 0.3845 | 0.3981 | 0.4524 | 0.5087 |

their empirical performance upper bound for these degradation groups, and the results are denoted as RealESRGAN-SG. As presented in Tab. 3, the fine-tuned models exceed the baseline models and obtain significant performance improvement on LPIPS. Although RealESRGAN-TG cannot surpass the empirical upper bound performance on LPIPS, it still achieves comparable performance. Moreover, our RealESRGAN-TG obtains higher PSNR compared to RealESRGAN-SG. This further shows the superiority of our method and suggests that multi-task learning enhances the performance of the specific task.

Table 4: Quantitative results of different methods on DIV2K8G. Group0 denotes the validation set with satisfactory degradation tasks, and Group1-7 represents the validation sets with unsatisfactory degradation tasks. The ESRGAN trained on the non-blind setting and RDSR trained on the MSE-based setting are marked in gray.

| | Metrics | ESRGAN | RDSR | BSRGAN | Real-SwinIR | DASR | MM-RealSR | RealESRGAN | RealESRGAN -TG (ours) |
|---|---|---|---|---|---|---|---|---|---|
| Group0 | PSNR (↑) | 21.39 | 25.12 | 24.19 | 23.61 | 23.97 | 23.61 | 24.08 | **24.10** |
| | LPIPS (↓) | 0.6251 | 0.5181 | 0.4567 | 0.4430 | 0.4697 | 0.4335 | 0.4297 | **0.4262** |
| Group1 | PSNR (↑) | 19.81 | 20.99 | 19.96 | 19.52 | 19.77 | 19.52 | 19.92 | **20.97** |
| | LPIPS (↓) | 0.6812 | 0.6325 | 0.6151 | 0.5954 | 0.6012 | 0.5572 | 0.5584 | **0.5249** |
| Group2 | PSNR (↑) | 21.10 | 22.55 | 21.91 | 21.16 | 21.77 | 21.39 | 21.67 | **22.72** |
| | LPIPS (↓) | 0.6636 | 0.5755 | 0.5091 | 0.4988 | 0.5175 | 0.4814 | 0.4823 | **0.4652** |
| Group3 | PSNR (↑) | 21.02 | 23.82 | 23.24 | 22.66 | 23.06 | 22.54 | 22.94 | **23.93** |
| | LPIPS (↓) | 0.6301 | 0.5225 | 0.4657 | 0.4528 | 0.4795 | 0.4445 | 0.4390 | **0.4188** |
| Group4 | PSNR (↑) | 22.10 | 24.64 | 23.96 | 23.37 | 24.05 | 23.20 | 23.63 | **24.41** |
| | LPIPS (↓) | 0.6221 | 0.5171 | 0.4617 | 0.4429 | 0.4596 | 0.4335 | 0.4294 | **0.4097** |
| Group5 | PSNR (↑) | 22.78 | 25.30 | 24.64 | 23.99 | 24.49 | 23.61 | 24.36 | **25.14** |
| | LPIPS (↓) | 0.5705 | 0.4873 | 0.4248 | 0.4113 | 0.4245 | 0.4072 | 0.4043 | **0.3821** |
| Group6 | PSNR (↑) | 22.42 | 25.32 | 24.58 | 23.87 | 24.52 | 23.71 | 24.38 | **24.86** |
| | LPIPS (↓) | 0.5849 | 0.4917 | 0.4255 | 0.4153 | 0.4269 | 0.4060 | 0.4039 | **0.3889** |
| Group7 | PSNR (↑) | 22.49 | 25.37 | 24.50 | 23.85 | 24.43 | 23.74 | 24.37 | **24.76** |
| | LPIPS (↓) | 0.5868 | 0.4958 | 0.4337 | 0.4227 | 0.4398 | 0.4110 | 0.4083 | **0.3976** |

**Impact of the number of task groups.** In Tab. 4, we group the unsatisfactory tasks into more compact new groups using new thresholds of [0.8, 0.7, 0.6, 0.5, 0.4, 0.3, 0.2], with the corresponding number of degradation tasks being [14, 15, 14, 31, 53, 116, 258]. The results demonstrate that our TGSR method exhibits steady improvement on the unsatisfactory tasks in Groups 1-7, while achieving comparable performance on the satisfactory tasks in Group 0. This indicates that our approach can be applied with a more fine-grained division of unsatisfactory tasks based on user requirements.

## 6 Conclusion

We have re-examined the real-SR problem through the lens of multi-task learning and introduced a novel approach, TGSR, to address task competition in real-SR. TGSR aims to identify and enhance the degradation tasks where a real-SR model underperforms. It involves a task grouping method and a simple fine-tuning approach using the identified task groups. Comprehensive evaluation confirms the effectiveness of our approach in producing high-quality super-resolution results in real-world scenarios. One potential limitation of our method is the need for adequate sampling from the degradation space.

## Acknowledgments and Disclosure of Funding

This work was supported in part by the National Natural Science Foundation of China under Grant (62276251,62272450), the Joint Lab of CAS-HK, the National Key RD Program of China (NO. 2022ZD0160100), and in part by the Youth Innovation Promotion Association of Chinese Academy of Sciences (No. 2020356).

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

# 7 Appendix

## 7.1 More Details of TGSR

In this section, we provide more training details of our TGSR. As mentioned in the main paper, we follow the RealESRGAN setting to train the real-SR network by standard GAN-based SR loss, including GAN loss $L_{GAN}$, L1 loss $L_1$, and perceptual loss $L_{per}$. The total loss can be defined as follows:

$$\mathcal{L} = w_1 \times \mathcal{L}_{L_1} + w_2 \times \mathcal{L}_{\text{per}} + w_3 \times \mathcal{L}_{\text{GAN}}, \tag{6}$$

The loss weights $w_1$, $w_2$, and $w_3$ are set to 1, 1, and 0.1, respectively. Specifically, for $\mathcal{L}_{L_1}$, we calculate the pixel loss as the $L_1$ distance $|\hat{y} - y|$, where $\hat{y}$ and $y$ denote the reconstructed HR image and the ground-truth HR image respectively. For the perceptual loss $\mathcal{L}_{\text{per}}$, we extract the conv1, conv2, conv3, conv4, conv5 feature maps of $\hat{y}$ and $y$ using the pre-trained VGG19 network [36], and the loss is calculated as the weighted sum of the respective $L_1$ distances between the feature maps of $\hat{y}$ and $y$, with weights set to [0.1, 0.1, 1, 1, 1] for each layer. For the adversarial loss $\mathcal{L}_{\text{GAN}}$, we use a U-Net discriminator with spectral normalization.

## 7.2 More Ablation Studies and Additional Experimental Results

### 7.2.1 Impact of the Increased Task Volume

In the main paper, we conducted experiments with a total of 4,000 (4k) degradation tasks. In this subsection, we have increased the number of tasks to 10,000 (10k) by sampling from the degradation space, in order to investigate the impact of the increased task volume on the results. In Fig. 9, we draw the histograms of the performance indicators computed with 4k (a) and 10k (b) degradation tasks respectively. It can be seen that their distributions are quite similar.

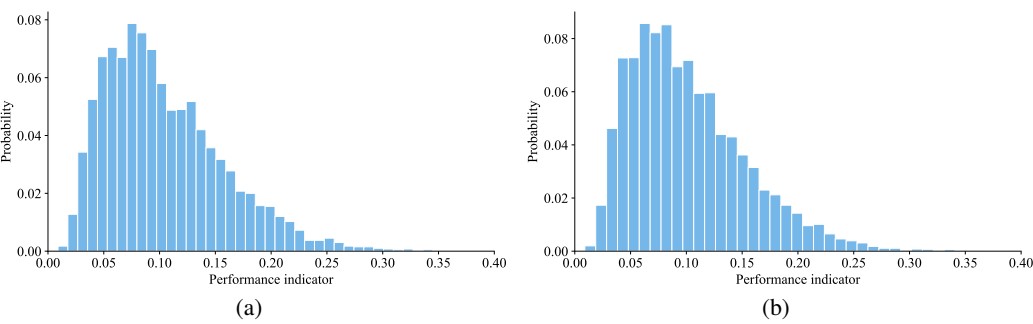

(a)                                      (b)

Figure 9: The histograms of the performance indicators with 4k (a) and 10k (b) degradation tasks respectively.

Furthermore, we use the same setup as in the main paper to do task grouping with the 10k degradation tasks and generate the test set for each group. We then compare the performance of TGSR trained with the 10k degradation tasks (TGSR_10k) and the 4k degradation tasks (TGSR_4k). In Tab. 5, we observe that the performance of TGSR_4k on the dataset DIV2K5G_10K is comparable to that of TGSR_10k. This finding suggests that increasing the number of tasks from 4,000 to 10,000 does not lead to significant improvements in performance for TGSR.

### 7.2.2 Necessity of Iterative Task Grouping

In Tab. 6, we provide the changes in performance improvement scores for various tasks across four training loops in the task grouping process in Alg. 1. It can be observed that many tasks show variations in their performance improvement scores across different training loops. For example, degradation task 5 exhibits a performance improvement of 0.57dB in the first training loop and 0.72dB in the third loop. This suggests that certain tasks are given priority during training in each loop, which leads to limited performance improvement for other tasks due to task competition. The observation supports the necessity of iterative task grouping as outlined in Alg. 1.

Table 5: Quantitative results of TGSR trained with the 4k and 10k degradation tasks, respectively. Group0 denotes the validation set with satisfactory degradation tasks, and Group1-4 represents the validation sets with unsatisfactory degradation tasks. The suffix 4k and 10k represent groups generated with the 4k and 10k degradation tasks, respectively.

| | Metrics | Group0_4k | Group1_4k | Group2_4k | Group3_4k | Group4_4k |
|---|---|---|---|---|---|---|
| RealESRGAN | PSNR (↑) | 23.85 | 20.11 | 22.08 | 24.00 | 24.59 |
| | LPIPS (↓) | 0.4325 | 0.5355 | 0.4701 | 0.4147 | 0.3970 |
| RealESRGAN-TG_4k | PSNR (↑) | 23.98 | 21.10 | 23.15 | 24.62 | 25.03 |
| | LPIPS (↓) | 0.4286 | 0.5056 | 0.4494 | 0.3975 | 0.3851 |
| RealESRGAN-TG_10k | PSNR (↑) | 23.91 | 21.03 | 22.98 | 24.50 | 24.97 |
| | LPIPS (↓) | 0.4299 | 0.5064 | 0.4525 | 0.4002 | 0.3859 |
| | Metrics | Group0_10k | Group1_10k | Group2_10k | Group3_10k | Group4_10k |
| RealESRGAN | PSNR (↑) | 23.81 | 19.85 | 20.95 | 23.77 | 24.38 |
| | LPIPS (↓) | 0.4324 | 0.5495 | 0.5073 | 0.4237 | 0.4019 |
| RealESRGAN-TG_4k | PSNR (↑) | 24.12 | 20.90 | 21.85 | 24.38 | 25.03 |
| | LPIPS (↓) | 0.4272 | 0.5187 | 0.4890 | 0.4046 | 0.3875 |
| RealESRGAN-TG_10k | PSNR (↑) | 24.05 | 20.94 | 21.85 | 24.31 | 24.93 |
| | LPIPS (↓) | 0.4285 | 0.5171 | 0.4907 | 0.4052 | 0.3886 |

Table 6: Performance improvement score of some degradation tasks across four training loops. NA means that the task is not available, since it has been grouped in the previous loop.

| Training loop | Deg.1 | Deg.2 | Deg.3 | Deg.4 | Deg.5 | Deg.6 | Deg.7 | Deg.8 | Deg.9 | Deg.10 |
|---|---|---|---|---|---|---|---|---|---|---|
| Train1 | 0.45 | 1.50 | 0.60 | 0.66 | 0.57 | 0.75 | 0.55 | 0.60 | 0.61 | 0.46 |
| Train2 | 0.47 | NA | 0.63 | 0.67 | 0.41 | 0.76 | 0.49 | 0.59 | 0.64 | 0.48 |
| Train3 | 0.48 | NA | NA | NA | 0.72 | NA | 0.65 | 0.63 | NA | 0.48 |
| Train4 | NA | NA | NA | NA | NA | NA | NA | NA | NA | NA |
| Degradation | Deg.11 | Deg.12 | Deg.13 | Deg.14 | Deg.15 | Deg.16 | Deg.17 | Deg.18 | Deg.19 | Deg.20 |
| Train1 | 0.43 | 0.60 | 0.47 | 0.27 | 0.66 | 0.53 | 0.22 | 0.31 | 0.63 | 0.20 |
| Train2 | 0.42 | 0.59 | 0.47 | 0.33 | 0.51 | 0.52 | 0.18 | 0.31 | 0.67 | 0.21 |
| Train3 | 0.48 | 0.63 | 0.50 | 0.31 | 0.73 | 0.54 | 0.18 | 0.37 | NA | 0.25 |
| Train4 | NA | NA | NA | 0.35 | NA | NA | 0.23 | 0.46 | NA | 0.26 |

### 7.2.3 Comparison with Single-task Network

It is commonly believed that a single-task network fine-tuned specifically for a particular degradation task can achieve the upper bound of performance for that task. However, in Fig. 10, we compare the PSNR improvement over RealESRGAN between our TGSR and the single-task network on a set of randomly selected tasks. Interestingly, we observe that the performance of the single-task network even decreases for tasks 1 and 4 in terms of PSNR, while our TGSR, which utilizes task grouping for multi-task learning, consistently improves performance across all tasks. On average, TGSR achieves a PSNR improvement of 0.74dB and an LPIPS improvement of 0.04, whereas the single-task network only achieves a PSNR improvement of 0.30dB and an LPIPS improvement of 0.03. This demonstrates the effectiveness of the task grouping and multi-task learning strategy in TGSR, which leverages the shared information across tasks to improve the generalization ability of the model and achieve better overall performance.

### 7.2.4 Impact of the Increased Volume of Unsatisfactory Tasks

Tab. 7 presents the results of TGSR with additional unsatisfactory tasks. We categorize the unsatisfied tasks into 9 groups based on thresholds [0.8, 0.7, 0.6, 0.5, 0.4, 0.3, 0.2, 0.15, 0.1] (0.15 and 0.1 are used to include more unsatisfactory tasks compared to Tab. 4), with the corresponding number of degradation tasks as [14, 15, 14, 31, 53, 116, 258, 190, 206]. We can observe that TGSR consistently achieves performance improvement on the unsatisfactory tasks in Groups 1-9 while maintaining comparable results on the satisfactory tasks in Group 0. It is worth noting that as the threshold for dividing the groups decreases, the performance gain becomes gradually smaller.

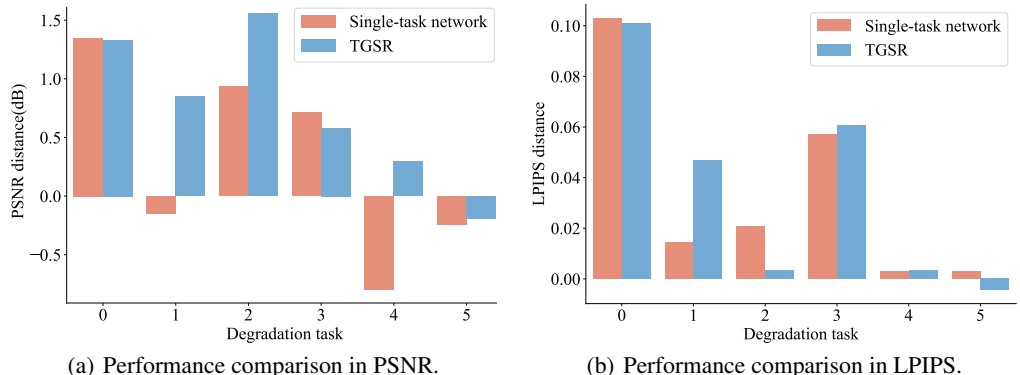

(a) Performance comparison in PSNR.  (b) Performance comparison in LPIPS.

Figure 10: Performance comparison of single-task network and TGSR in (a) PSNR and (b) LPIPS.

Table 7: Quantitative results of different methods on DIV2K10G. Group0 denotes the validation set with satisfactory degradation tasks, and Group1-9 represents the validation sets with unsatisfactory degradation tasks.

|  | Metrics | ESRGAN | RDSR | BSRGAN | RealSwinIR | DASR | MM-RealSR | RealESRGAN | RealESRGAN-TG (ours) |
|---|---|---|---|---|---|---|---|---|---|
| Group0 | PSNR (↑) | 21.15 | 24.80 | 23.66 | 22.97 | 23.66 | 23.28 | 23.65 | **23.78** |
|  | LPIPS (↓) | 0.6268 | 0.5255 | 0.4616 | 0.4503 | 0.4743 | 0.4403 | 0.4358 | **0.4344** |
| Group1 | PSNR (↑) | 20.01 | 21.13 | 20.07 | 19.62 | 19.85 | 19.63 | 20.01 | **21.07** |
|  | LPIPS (↓) | 0.6708 | 0.6267 | 0.6051 | 0.5834 | 0.5923 | 0.5552 | 0.5563 | **0.5188** |
| Group2 | PSNR (↑) | 21.08 | 22.38 | 21.82 | 20.97 | 21.53 | 21.06 | 21.36 | **22.47** |
|  | LPIPS (↓) | 0.6687 | 0.5821 | 0.5177 | 0.5092 | 0.5200 | 0.4910 | 0.4922 | **0.4735** |
| Group3 | PSNR (↑) | 20.94 | 23.58 | 23.06 | 22.40 | 22.82 | 22.30 | 22.72 | **23.58** |
|  | LPIPS (↓) | 0.6421 | 0.5410 | 0.4809 | 0.4671 | 0.4940 | 0.4564 | 0.4553 | **0.4393** |
| Group4 | PSNR (↑) | 22.10 | 24.80 | 24.27 | 23.56 | 24.31 | 23.46 | 23.82 | **24.40** |
|  | LPIPS (↓) | 0.6098 | 0.5082 | 0.4470 | 0.4326 | 0.4496 | 0.4257 | 0.4238 | **0.4047** |
| Group5 | PSNR (↑) | 22.84 | 25.58 | 24.89 | 24.27 | 24.91 | 23.85 | 24.56 | **25.07** |
|  | LPIPS (↓) | 0.5626 | 0.4843 | 0.4208 | 0.4057 | 0.4187 | 0.4009 | 0.3974 | **0.3768** |
| Group6 | PSNR (↑) | 22.50 | 25.11 | 24.39 | 23.72 | 24.42 | 23.64 | 24.10 | **24.63** |
|  | LPIPS (↓) | 0.5895 | 0.4979 | 0.4400 | 0.4254 | 0.4410 | 0.4161 | 0.4130 | **0.3993** |
| Group7 | PSNR (↑) | 22.41 | 25.54 | 24.72 | 24.03 | 24.44 | 23.80 | 24.50 | **24.90** |
|  | LPIPS (↓) | 0.5884 | 0.4892 | 0.4248 | 0.4146 | 0.4341 | 0.4048 | 0.4016 | **0.3897** |
| Group8 | PSNR (↑) | 20.94 | 25.08 | 24.19 | 23.57 | 23.77 | 23.44 | 24.10 | **24.52** |
|  | LPIPS (↓) | 0.6223 | 0.5057 | 0.4492 | 0.4354 | 0.4568 | 0.4226 | 0.4211 | **0.4112** |
| Group9 | PSNR (↑) | 19.86 | 24.89 | 23.85 | 23.29 | 23.65 | 23.44 | 23.97 | **24.29** |
|  | LPIPS (↓) | 0.6340 | 0.5140 | 0.4624 | 0.4485 | 0.4820 | 0.4333 | 0.4302 | **0.4209** |

## 7.3 More Visual Results

Fig. 11 presents additional sample images from our validation datasets, including some unsatisfactory degradation tasks from Groups 1-4 and some satisfactory degradation tasks from Group 0. These images are accompanied by the corresponding ground-truth images. Fig. 12 presents an additional qualitative comparison of competing methods on DIV2K5G. Fig. 13 presents an additional qualitative comparison of competing methods on real-world images. Through the grouping and handling of unsatisfactory degradation tasks, our TGSR exhibits enhanced generalization ability when applied to real-world images.

### 7.3.1 More Sample Images in DIV2K5G

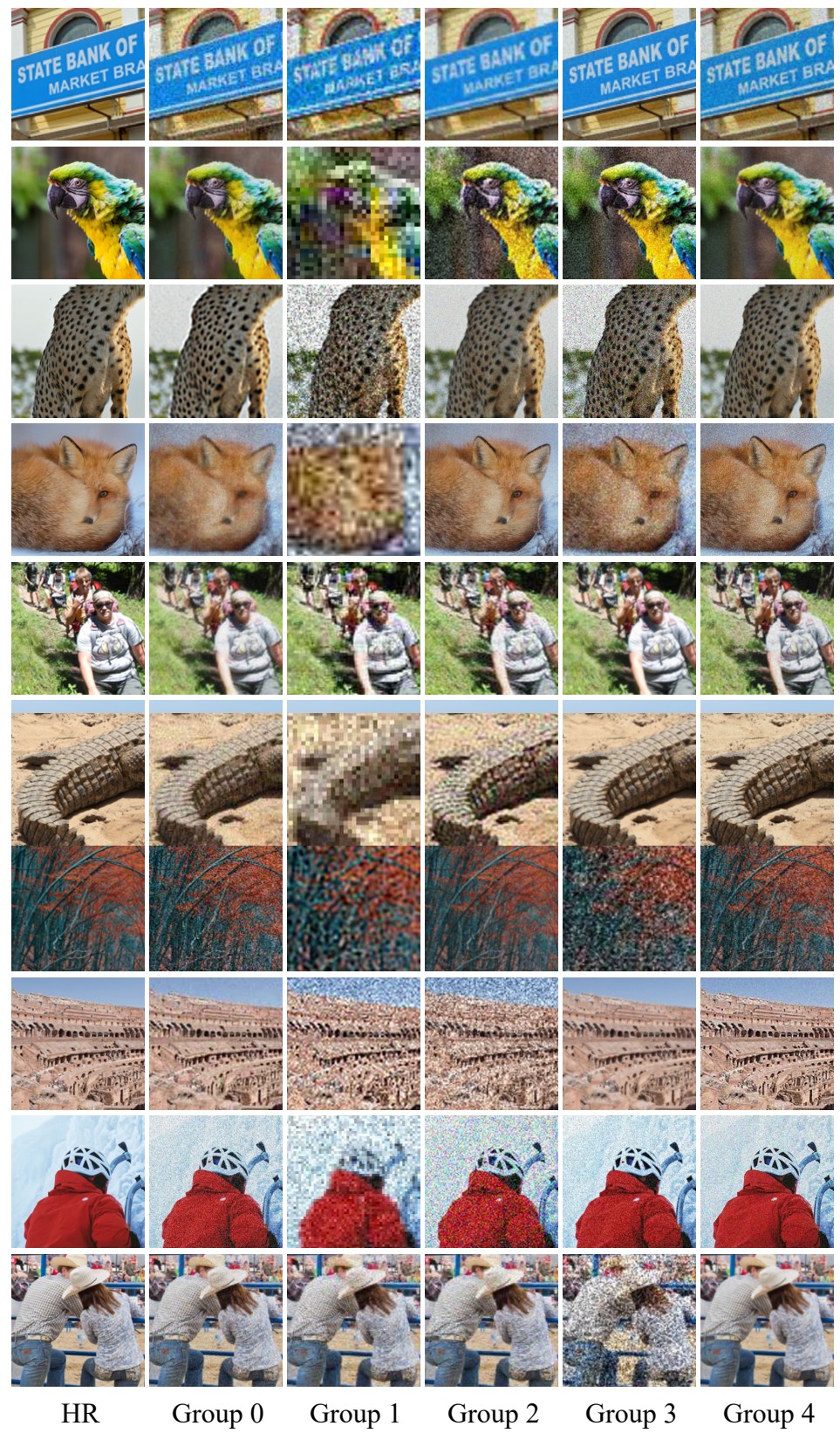

HR    Group 0    Group 1    Group 2    Group 3    Group 4

Figure 11: Sample images from different degradation task groups in our DIV2K5G datasets.

### 7.3.2 More Qualitative Comparisons on DIV2K5G

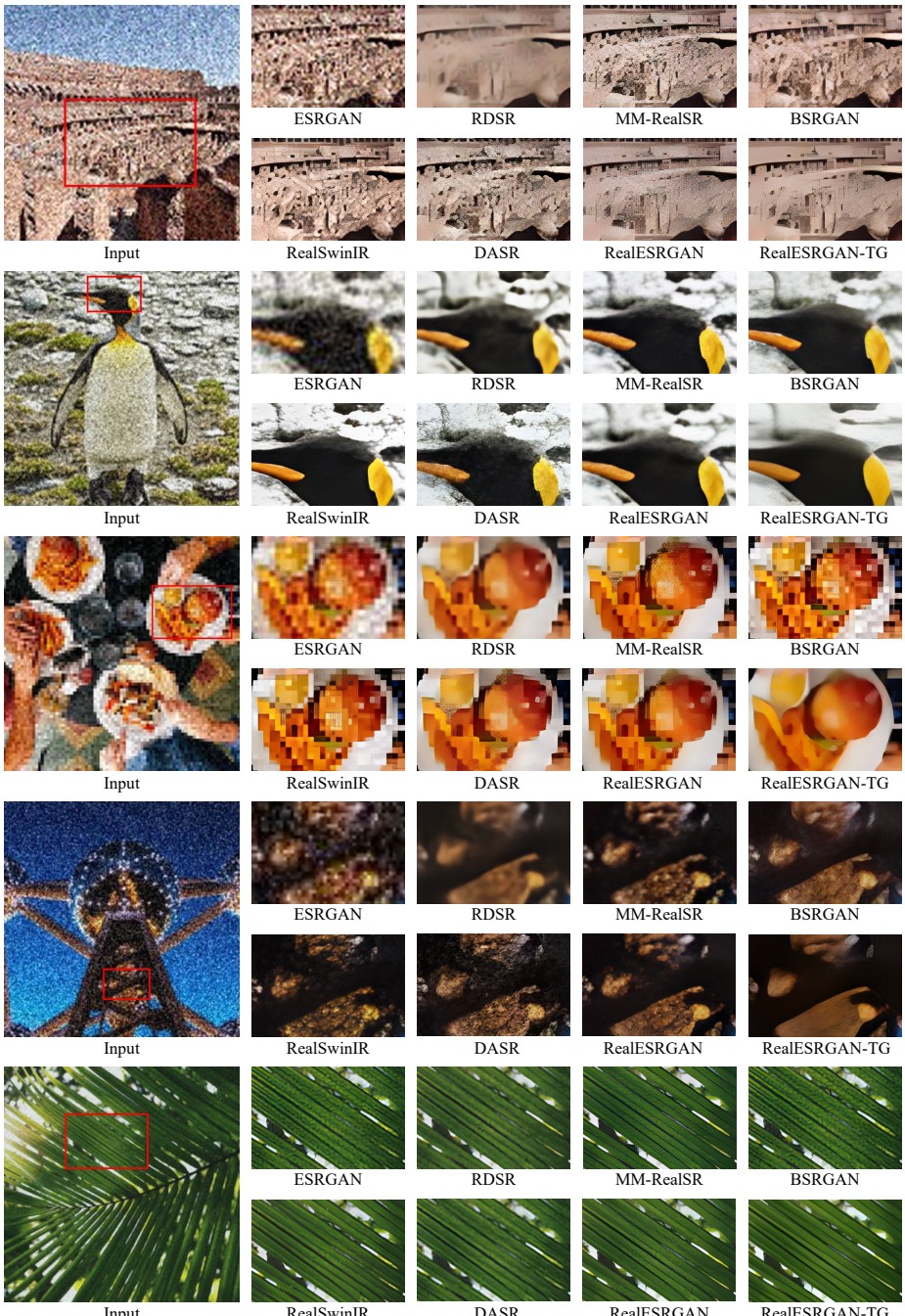

Figure 12: Qualitative comparison of competing methods on DIV2K5G.

### 7.3.3 More Qualitative Comparisons on Real-world Images

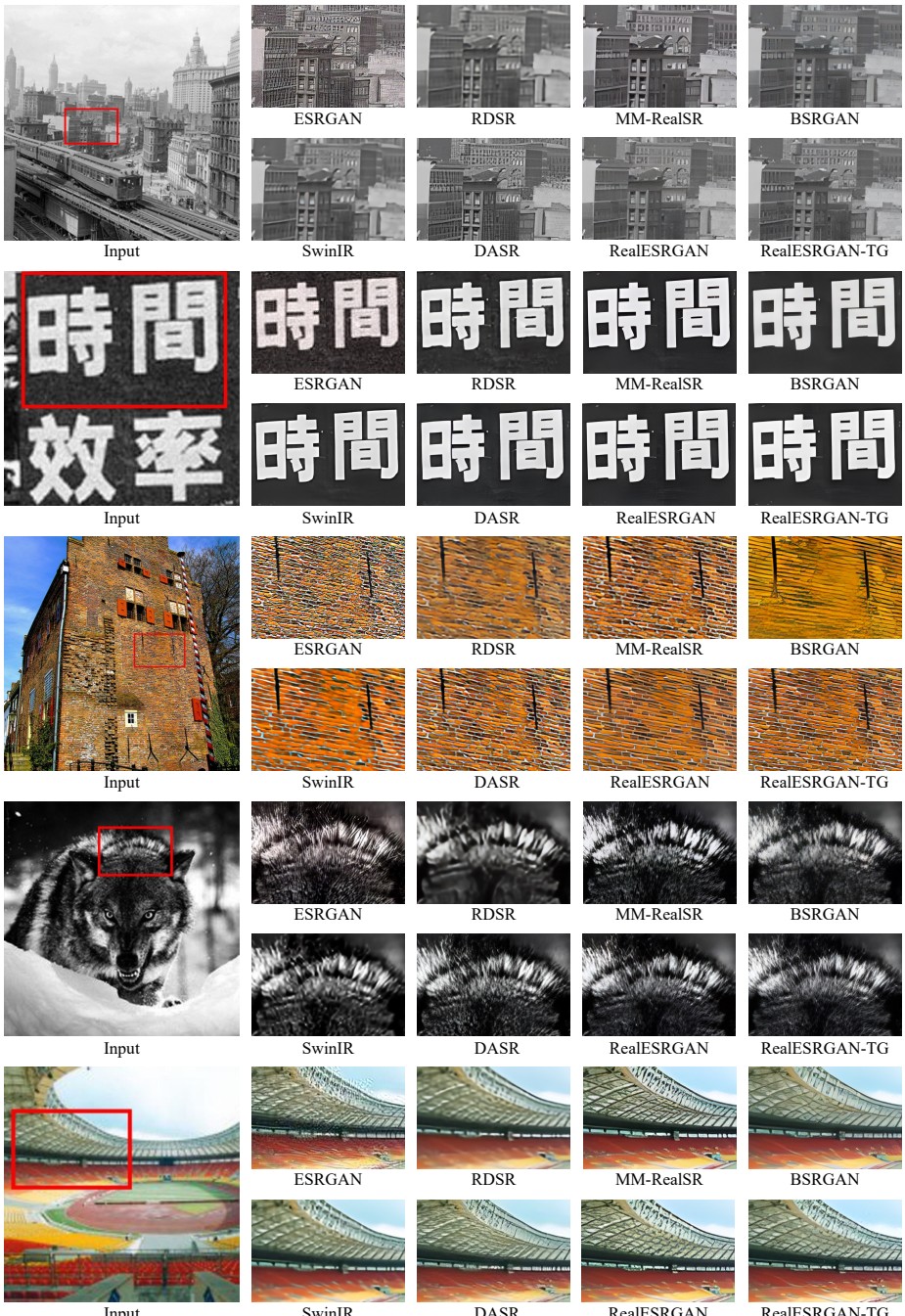

Figure 13: Qualitative comparison of competing methods on real-world images.

