# OpenReview forum: "Real-World Image Super-Resolution as Multi-Task Learning"
_NeurIPS.cc/2023/Conference — NeurIPS 2023 poster_

### Official Review · Reviewer_o9KK · 2023-07-04

**Soundness:** 3 good
**Presentation:** 3 good
**Contribution:** 3 good
**Rating:** 5
**Confidence:** 4

**Summary:**

This paper revisits the real-world super-resolution task from the perspective of multi-task learning, considering each type of degradation as a separate task. However, there are countless types of degradation in the real world, which often results in severe task competition in previous methods. The authors propose a task grouping strategy to address this problem.
Furthermore, they present a multi-task learning framework suitable for real SR tasks, guiding the target SR model by the task group label probability. Experimental results showcase an advanced performance improvement by the proposed TGSR across a wide range of degradation types.


**Strengths:**

This paper addresses real-world super-resolution tasks from the multi-task learning perspective, which I think is novel to this field. The paper effectively dissects the issues present in previous methods from a multi-task learning perspective and provide in-depth insight.

I think that the degradation task grouping strategy proposed in the paper may sheds new light on the analysis and study of the blind SR tasks.

**Weaknesses:**

Detailed selection strategy for different degradation tasks in Section 3.1 and Figure 1 are recommended.

Hyper-parameters in the proposed task grouping strategy, such as the number of task group and division threshold, are given and fixed in this paper, which is not motivated to me. I think these could lead to potential impact for the proposed method, but detailed analysis or ablation is missing is this paper. The authors should provide more comprehensive analyses to address these concerns.

In the degradation type grouping strategy, the threshold for each group division is determined manually, and the paper does not mention ablation in the number of groups or threshold, which could lead to potential imprecision in the division strategy.


**Questions:**

My concerns regarding the evaluation data DIV2K5G and its close connection with the proposed TGSR model are valid. The potential for an inherent bias exists since the dataset was constructed using the proposed task grouping algorithm, which is closely tied with the iteratively trained and fine-tuned SR model.

It would indeed be beneficial for the authors to evaluate the TGSR model on the standard benchmark dataset, which would provide a more independent measure of the effectiveness of the multi-task learning-based task grouping strategy.

Additionally, performing comparisons on the RealSR dataset or a randomly generated benchmark by BSRGAN or RealESRGAN can be beneficial. These additional evaluations can provide further evidence to the superiority of the proposed TGSR.


**Limitations:**

None.

---

> ### Author Rebuttal · Authors · 2023-08-10
>
>
> Dear Reviewer o9KK,
>
> We appreciate your thoughtful comments. Please find below our detailed response addressing your concerns.
>
> ### Q1. Detailed selection strategy for different degradation tasks in Section 3.1 and Figure 1 are recommended.
>
> Please note that we've stated in **Line 115** that the 100 degradation tasks (used for the illustration in Figure 1) were randomly sampled from the degradation model of RealESRGAN. In our main experiment, we also randomly generated 4000 degradation tasks using the RealESRGAN degradation model.
>
> ### Q2. Hyper-parameters in the proposed task grouping strategy, such as the number of task group and division threshold, are given and fixed in this paper, which is not motivated to me. I think these could lead to potential impact for the proposed method, but detailed analysis or ablation is missing is this paper. The authors should provide more comprehensive analyses to address these concerns.
>
> Please note that in **Section 7.3.4** of the supplementary materials, we've provided additional results and a discussion about the effect of the number of task groups. We find that as more and more groups are added, the rate of performance improvement for each new group becomes smaller.
>
> ### Q3. In the degradation type grouping strategy, the threshold for each group division is determined manually, and the paper does not mention ablation in the number of groups or threshold, which could lead to potential imprecision in the division strategy.
>
> Please note that in **Section 7.3.5** of the supplementary materials, we've provided an ablation study of the division threshold. We find that as the division threshold decreases, the performance gain becomes smaller.
>
> ### Q4. Performing comparisons on the RealSR dataset or a randomly generated benchmark by BSRGAN or RealESRGAN can be beneficial. These additional evaluations can provide further evidence to the superiority of the proposed TGSR.
>
> Thank you for the suggestion. Following the approach used by BSRGAN, we generate a new dataset DIV2K\_random by randomly adding degradations sampled from the degradation model of RealESRGAN to the DIV2K\_val dataset. Additionally, we conduct a comparison on the real benchmark AIM2019-val, the test set used for the real SR track in the AIM 2019 Challenge [1]. On both datasets, our TGSR outperforms state-of-the-art methods consistently and significantly. We will include the results in the final version.
>
>
> |          |        | ESRGAN|BSRGAN|RealESRGAN|SwinIR|DASR|TGSR|
> |  ----  | ----  |  ----  | ----  |  ----  | ----  |  ----  | ----  |
> | DIV2K\_random| PSNR |20.63 |23.76|23.54|23.13|23.52|**23.84**|
> | DIV2K\_random| LPIPS |0.6345 | 0.4622|0.4423|0.4432|0.4832|**0.4368**|
> | AIM2019-val | PSNR | 23.16 | 24.20 | 23.89 | 23.89 | 23.76 | **24.27**  |
> | AIM2019-val | LPIPS | 0.5500 | 0.4000 | 0.3960 | **0.3870** | 0.4210 | **0.3899** |
>
> [1] AIM 2019 challenge on real-world image super-resolution: Methods and results.

---

> > ### Comment · Reviewer_o9KK · 2023-08-19
> >
> > I have carefully read author's further feedback for all reviewers, and the additional information provided has address my concerns. However, echoing other reviewers' sentiments, the current manuscript falls short in terms of in-depth analysis and sufficient verification. There's definite potential for improvement in forthcoming revisions, especially with regards to intuitive observations, underlying motivations, and comprehensive comparative and ablation studies. A more rigorous refinement is pivotal for the manuscript to resonate with the stringent standards of NeurIPS. After careful consideration, I've opted to maintain my initial evaluation score.

---

> > > ### Author Response · Authors · 2023-08-21
> > >
> > > We appreciate your response and will include more details and your suggestion in our camera-ready version.

---

### Official Review · Reviewer_yhLt · 2023-07-05

**Soundness:** 4 excellent
**Presentation:** 3 good
**Contribution:** 4 excellent
**Rating:** 8
**Confidence:** 5

**Summary:**

The authors rethink the real-world super-resolution problem from the perspective of multi-task learning. And point out the primary challenge: task competition problem. To address this issue, they propose a task grouping method to identify unsatisfactory tasks and introduce TGSR to handle them separately, thereby eliminating task competition. Extensive experiments demonstrate the effectiveness of the proposed method.

**Strengths:**

1.	Taking real-SR as a multi-task learning problem offers a novel perspective that can yield further insights and considerations. For instance, how to effectively sample degradation in a large degradation space? What relationships exist between different tasks?

2.	The authors present a clear motivation in the form of task competition, as current real-SR networks are unable to perfectly handle all cases of degradation.

3.	The paper includes extensive ablation studies. Interestingly, the authors found that even when fine-tuning directly on a single degradation task, many cases still do not perform as well as a range of degradation. This suggests that traditional blind super-resolution networks based on kernel estimation may no longer be applicable within large degradation spaces. This is a highly insightful finding.


**Weaknesses:**

1.	The authors should provide the calculation time of the performance indicator, although the performance indicator is obviously faster than directly fine-tuning a single-task network.

2.	The groups 1-4 in Figure 4 of the main text seem to be inconsistent with the samples in the supp file, and Table 1 also shows that group 1 should be a low-quality difficult case (with the lowest PSNR value).


**Questions:**

See above weakness, authors need to address them well.

**Limitations:**

Yes. Besides, since this work still uses synthetic data. It would be better to consider low quality images from real scenes.

---

> ### Author Rebuttal · Authors · 2023-08-10
>
>
> Dear Reviewer yhLt,
>
> We'd like to thank you for your positive feedback and address the concerns raised in your comments.
>
> ### Q1. The authors should provide the calculation time of the performance indicator, although the performance indicator is obviously faster than directly fine-tuning a single-task network.
>
> The computational time of the performance indicator for one degradation task is around 100 seconds. In contrast, it takes about 2 hours to fine-tune a single-task network. Hence, our approach offers a nearly 70-fold speed increase when compared to the direct fine-tuning of single-task networks for identifying unsatisfactory tasks. We will include the comparison and discussion of this matter in the final version.
>
> ### Q2. The groups 1-4 in Figure 4 of the main text seem to be inconsistent with the samples in the supp file, and Table 1 also shows that group 1 should be a low-quality difficult case (with the lowest PSNR value).
>
> Thank you for your careful reading. The order of groups 1-4 in Figure 4 has been incorrectly reversed, but the order in the supplement is correct. We will correct this error in the final version.

---

> > ### Comment · Reviewer_yhLt · 2023-08-13
> >
> > This response has resolved my concerns. From my perspective, this paper highlights the task competition in real-SR problem and provide a reasonable solution. Overall, there's no work that has explored real SR from a multi-task view,  so I think that this paper is insightful and can inspire further research due to this theoretical contribution.
> >
> > I have also read other reviews and the corresponding feedback, and although there may exist some small problems in this paper, the overall insight of this paper is significant, and the feedback also gives me some inspiration. I suggest authors add them in the revision.
> >
> > Since all of my concerns have been solved and I think other reviews would not affect its contribution, I would like to raise my score.

---

> > > ### Author Response · Authors · 2023-08-21
> > >
> > > We appreciate your feedback and suggestions. We will incorporate them into the final version.
> > > We also appreciate your constructive discussion with Reviewer 4Sx7.

---

### Official Review · Reviewer_4Sx7 · 2023-07-05

**Soundness:** 2 fair
**Presentation:** 2 fair
**Contribution:** 2 fair
**Rating:** 5
**Confidence:** 4

**Summary:**

This paper aims at the task conflict issue of real-world image super-resolution (SR) with multiple degradation tasks, and proposes a task grouping approach to group similar tasks together to mitigate task competition. In addition, this paper designs a real-SR network called TGSR (task grouping-based real-SR network), which leverages the identified task groups to train a task group classifier and use the predicted information to generate modulation signals for image restoration.

**Strengths:**

1. a new look at real SR from a multi-task learning perspective.
2. a performance indicator based on gradient updates to efficiently identify the degradation tasks.

**Weaknesses:**

1. The authors compare the PSNR distance between the real-SR model trained on entire degradation space introduced by Real-ESRGAN [34] and 100 types of single-task models fine-tuned on the specific-degradation, and highlight some degradation tasks not well solved by the real-SR model as unsatisfactory tasks. However, they only number 1-100 for different tasks, and do not present the detailed settings (blur type, kernel width, noise type, noise level, and jpeg compression) of each degradation task. Such inadequate expression makes readers including me confued as to whether there is a relationship between PSNR distance and degradation type.

2. The authors claim that "... divide them (unsatisfactory degradation tasks) into multiple task groups based on their similarity to reduce negative transfer." However, the task similarity computed by the PSNR distance is not convincing, and it is worth to further studying the relationship between the task grouping strategy and the degradation distribution.

3. At lines 177-178, "in this study, we use the found degradation task groups to train a task group classifier, which is then integrated into a standard real-SR framework". I want to know how to select the number of task groups, and how do different numbers of groupings affect the final performance. If the number of groupings is set to infinity, is the task group classifier equivalent to the well-studied degradation estimator?

**Questions:**

1. AT lines 166-167, "we make a trade-off to fine-tune the pre-trained real-SR network on all unsatisfactory tasks simultaneously through joint-training". What is the joint learning in this paper?


2. It is unfair to conduct the evaluation experiments on the grouped DIV2k (namely, DIV2K5G) with 5 different degradation groups. The authors should also conduct a fair comparison with Sota methods on the well-used benchmarks, such as SwinIR [A1], DASR [A2].

[A1] Liang J, Cao J, Sun G, et al. Swinir: Image restoration using swin transformer[C]//Proceedings of the IEEE/CVF international conference on computer vision. 2021: 1833-1844.
[A2] Wang L, Wang Y, Dong X, et al. Unsupervised degradation representation learning for blind super-resolution[C]//Proceedings of the IEEE/CVF Conference on Computer Vision and Pattern Recognition. 2021: 10581-10590.

---

> ### Author Rebuttal · Authors · 2023-08-10
>
>
> Dear Reviewer 4Sx7,
>
> We appreciate your feedback and would like to address the concerns raised in your comments, which include some factual errors and misunderstandings.
>
> ## Factual Errors & Misunderstandings
>
> ### Q1. The authors compare the PSNR distance... However, they only number 1-100 for different tasks, and do not present the detailed settings (blur type, kernel width, noise type, noise level, and jpeg compression) of each degradation task. Such inadequate expression makes readers including me confued as to whether there is a relationship between PSNR distance and degradation type.
>
> (1) Please note that **the 100 degradation tasks are only used for illustration in Figure 1**, as stated in **Line 115**.
>
> (2) Please note that we sampled **4,000** tasks for the main experiments (**Line 198**) and **10,000** tasks for the experiments in the supplementary materials (**Line 460**), which can sufficiently represent the degradation space. To efficiently identify the unsatisfactory tasks from such a large number of tasks, we do not train a single-task network for each task, but propose a gradient-based performance indicator as described in **Lines 138-148**.
>
> (3) Please also note that due to the complex combinations of degradations in a large degradation space (e.g., RealESRGAN degradation model), it is very difficult to group degradation tasks by their types, as pointed out in [1].
>
> ### Q2. The authors claim that "... divide them (unsatisfactory degradation tasks) into multiple task groups based on their similarity to reduce negative transfer." However, the task similarity computed by the PSNR distance is not convincing, and it is worth to further studying the relationship between the task grouping strategy and the degradation distribution.
>
> (1) The task grouping approach (e.g., the CVPR-2018 best paper: Taskonomy [2]) widely used in multi-task learning [2,3,4] focuses on finding the tasks that should be trained together based on the performance differences (e.g., PSNR distance) between the multi-task network and single-task networks.
>
> (2) As mentioned in [4], the term “task similarity” can easily be misunderstood to imply a strong attribute relationship between tasks. In fact, the task similarity in our paper represents the affinity relationship between the single-task network and the multi-task (real-SR) network, rather than the similarity between degradation types. We will further clarify this in the final version.
>
> (3) The consensus in the field recognizes that improved performance, such as a higher PSNR, contributes to superior visual outcomes. Our experimental findings further validate the achievement of state-of-the-art results.
>
> ## Other Comments
>
> ### Q3. At lines 177-178, "in this study, we use the found degradation task groups to train a task group classifier, which is then integrated into a standard real-SR framework". I want to know how to select the number of task groups, and how do different numbers of groupings affect the final performance. If the number of groupings is set to infinity, is the task group classifier equivalent to the well-studied degradation estimator?
>
> Please note that in **Section 7.3.4 (Line 490)** of the supplementary materials, we've provided additional results and a discussion about the effect of the number of task groups. The number of task groups is determined by the threshold (**Line 206**) of performance improvement. We find that as more and more groups are added, the rate of performance improvement for each new group becomes smaller. Our approach differs from the degradation estimation method, which estimates every degradation task, because we have a limited group number of unsatisfactory tasks.
>
> ### Q4. AT lines 166-167, "we make a trade-off to fine-tune the pre-trained real-SR network on all unsatisfactory tasks simultaneously through joint-training". What is the joint learning in this paper?
>
> Joint-training refers to training a single real-SR network with multiple (e.g., thousands of) degradation tasks simultaneously, which is the strategy used by most existing real-SR methods such as RealESRGAN and BSRGAN, as stated in **Lines 139-140**.
>
> ### Q5. It is unfair to conduct the evaluation experiments on the grouped DIV2k (namely, DIV2K5G) with 5 different degradation groups. The authors should also conduct a fair comparison with Sota methods on the well-used benchmarks, such as SwinIR [A1], DASR [A2].
>
> Thank you for the suggestion. Following the approach used by SwinIR and BSRGAN, we generate a new dataset DIV2K\_random by randomly adding degradations sampled from the degradation model of RealESRGAN to the DIV2K\_val dataset. Additionally, we conduct a comparison on the real benchmark AIM2019-val, the test set used for the real SR track in the AIM 2019 Challenge [5]. On both datasets, our TGSR outperforms state-of-the-art methods consistently and significantly. We will include the results in the final version.
>
> |          |        | ESRGAN|BSRGAN|RealESRGAN|SwinIR|DASR|TGSR|
> |  ----  | ----  |  ----  | ----  |  ----  | ----  |  ----  | ----  |
> | DIV2K\_random| PSNR |20.63 |23.76|23.54|23.13|23.52|**23.84**|
> | DIV2K\_random| LPIPS |0.6345 | 0.4622|0.4423|0.4432|0.4832|**0.4368**|
> | AIM2019-val | PSNR | 23.16 | 24.20 | 23.89 | 23.89 | 23.76 | **24.27**  |
> | AIM2019-val | LPIPS | 0.5500 | 0.4000 | 0.3960 | **0.3870** | 0.4210 | **0.3899** |
>
> [1] Crafting Training Degradation Distribution for the Accuracy-Generalization Trade-off in Real-World Super-Resolution. ICML2023.
>
> [2] Taskonomy: Disentangling task transfer learning
>
> [3] Conflict-averse gradient descent for multi-task learning
>
> [4] Efficiently Identifying Task Groupings for Multi-Task Learning
>
> [5] AIM 2019 challenge on real-world image super-resolution: Methods and results

---

> > ### Comment · Reviewer_4Sx7 · 2023-08-19
> >
> > Thanks for the rebuttal and it has addressed some of my concerns. In my opinion, the group classifier works like a degradation classifier that identifies LR images with specific degradations, which can be validated in Fig. 4. Instead of learning degradation representations that distinguish different degradation types [33], this classifier simplify the degradation representation learning task to a classification task on a set of groups. Still, I have several concerns:
> >
> > (1) I wonder the contribution of the CE term in Eq. 6. Without the CE loss term, the classifier is opimized according to the SR loss and I wonder whether this can produce better performance than the heuteristic approach used in this paper for grouping.
> > (2) I agree with Reviewer LkK8 that the performance distance may be caused by many factors and this indicator seems not very convincing to measure the task competition.

---

> > > ### Author Response · Authors · 2023-08-19
> > >
> > > #### 1. I wonder the contribution of the CE term in Eq. 6. Without the CE loss term, the classifier is opimized according to the SR loss and I wonder whether this can produce better performance than the heuteristic approach used in this paper for grouping.
> > >
> > > Thank you for your question. Following your suggestion, we've conducted the experiment and the results are presented in the table below. The results clearly show that TGSR w/o CE, which optimizes the SR network soley with the SR loss, only exhibits slight improvement over RealESRGAN on groups 0, 2, and 4, whereas it performs considerably worse than our proposed TGSR. This serves as validation for the effectiveness of our task grouping method, wherein we identify underperforming tasks with comparable performance levels and train them collectively. It highlights the positive impact of training similar tasks together, aligning with previous findings in the multi-task learning literature.
> > >
> > >
> > > |            |            | Group0 | Group1 | Group2 |Group3 |Group4 |
> > > |------      |------      | ------- | ------- |------- |------- |------- |
> > > | RealESRGAN | PSNR       |  23.85 |  20.10        |   22.07    |    24.30    |  24.58      |
> > > |            | LPIPS      |  0.4325| 0.5355        | 0.4701      | 0.4147      | 0.3970      |
> > > | TGSR w/o CE| PSNR       |  23.94 | 20.08        |  22.16     |    24.17   | 24.75      |
> > > |            | LPIPS      |  0.4303  | 0.5348         |  0.4696     | 0.4155      |   0.3963    |
> > > | TGSR       | PSNR       | **23.99** |   **21.10**      |  **23.15**     |  **24.62**     |  **25.03**     |
> > > |            | LPIPS      | **0.4286** |  **0.5056**      | **0.4494**      |  **0.3975**     | **0.3851**      |
> > >
> > > #### 2. I agree with Reviewer LkK8 that the performance distance may be caused by many factors and this indicator seems not very convincing to measure the task competition.
> > >
> > > First, please note that *performance indicators that leverage a comparison between the multi-task network and single-task networks have been extensively employed in the field of multi-task learning* (e.g., the quality rate in [1] and the relative performance indicator in [2] and [3]). Our proposed performance indicator follows the same design principle.
> > >
> > > Second, as stated in **Lines 114-124**, the only variable we adjusted is the number of degradation tasks. Therefore, the performance drop can be soley attributed to the increased number of tasks. This observation aligns with the principles of multi-task learning [1,2,3], where the phenomenon is commonly referred to as task competition, as also mentioned in **Lines 111-114**.
> > >
> > > [1] Zamir A R, Sax A, Shen W, et al. Taskonomy: Disentangling task transfer learning[C]//Proceedings of the IEEE conference on computer vision and pattern recognition. 2018: 3712-3722.
> > >
> > > [2] Fifty, C., Amid, E., Zhao, Z., Yu, T., Anil, R., & Finn, C. (2021). Efficiently identifying task groupings for multi-task learning. Advances in Neural Information Processing Systems, 34, 27503-27516.
> > >
> > > [3] Standley, T., Zamir, A., Chen, D., Guibas, L., Malik, J., & Savarese, S. (2020, November). Which tasks should be learned together in multi-task learning?. In International Conference on Machine Learning (pp. 9120-9132). PMLR.

---

> > > > ### Comment · Reviewer_4Sx7 · 2023-08-21
> > > >
> > > > Thanks for your response. I am happy to see the ablation results regarding the CE loss and impressed about the gains. I think these results should be included to better demonstrate the effectiveness of the idea of hard sample grouping. I am not very familiar with multi-task learning and my concerns regarding the metrics still remain. However, as highlighted by Reviewer yhLt, this paper makes the first attempt and may shed some lights on the following methods. Overall, I decide to change my rating to BA.

---

> > > > > ### Author Response · Authors · 2023-08-21
> > > > >
> > > > > Thank you very much for your efforts in the review process.
> > > > > In the final version, we will include the ablation results, clarify the description of the metric, and incorporate all suggestions. Thank you again for your discussion.

---

> > > ### Comment · Reviewer_yhLt · 2023-08-19
> > >
> > > Thanks for your comment, I would like to express different opinions here because I think the design of the distance metric is reasonable.
> > >
> > > Since I have previously studied all-in-one image restoration task that is related to multi-task, I am familiar with multi-task learning, such as taskonomy, PCGrad, CAGrad, and MMoE, etc. In my opinion, this paper's design of performance distance satisfies multi-task learning, and there's no work that explores how the multi-task would affect the learning process in low-level vision areas as I know. Therefore, as a first step, it is reasonable to follow the setting of multi-task learning to set a metric for this multiple degradation restoration task, and other potential metrics can be future work since there's no work that has explored this issue before. After all, I believe that it would not be good to be too harsh on a new thing.

---

### Official Review · Reviewer_tGSk · 2023-07-05

**Soundness:** 2 fair
**Presentation:** 2 fair
**Contribution:** 2 fair
**Rating:** 3
**Confidence:** 5

**Summary:**

This paper models a real-world image super-resolution (real-SR) from a multi-task learning perspective, that is treat real-SR as solving multiple distinct degradation tasks. To this end, the authors propose a task-grouping approach by grouping similar tasks together. Extensive experiments demonstrate the effectiveness of the proposed method.

**Strengths:**

The authors regard learning a real-SR model as a multi-task learning task and highlight the task competition problem. The authors develop a task grouping-based real-SR network (TGSR).

**Weaknesses:**

1. The novelty of this paper should be highlighted. This paper refers to the tasks that are solved by the real-SR network well or not as unsatisfactory and satisfactory tasks. Such a way is similar to [1] which classifies the images as simple, medium and hard. It would be better to highlight the novelty of the method.
    [1] ClassSR: A General Framework to Accelerate Super-Resolution Networks by Data Characteristic

2. The multi-task Real-SR method seems to be not smart. There are infinite number of degradations. The method randomly samples 100 degradation tasks and obtains 100 fine-tuned models.

3. The task-grouping approach groups similar tasks together. The similarity is based on the performance. It means that the low noisy degradation and weak blurry degradation are similar. However, low noise and strong noise are the same degradations, which should be treated as the same task.

4. In Figure 1 (a), this paper uses the degradation models which are based on Real-ESRGAN. Each degradation on an image is a composite function which different degradations. Could you investigate the effect of such a case?

5. The authors identify the unsatisfactory tasks by comparing the performance between the jointly-trained multi-task real-SR network and the fine-tuned single-task networks. What if directly compare the PSRN (like [1]) with a threshold?

6. Some experiment details are not clear. Is the real-SR model fine-tuned on a pre-trained Real-ESRGAN or trained from scratch?

7. The performance is not state-of-the-art. In Table 1, the proposed method is worse than RDSR. In Table 2, RealESRGAN-G has no advantage over RealESRGAN for Group0 and Group1. How to demonstrate the superiority of the proposed method. In Figure 5, the input images have much noise, not low-resolution. It would be better to provide SR results.

**Questions:**

Please refer to the Weaknesses.



**Limitations:**

The authors do not address the limitations in the paper. It would be better to include the above suggestions for improvement.

---

> ### Author Rebuttal · Authors · 2023-08-10
>
>
> Dear Reviewer tGSk,
>
> Thank you for your feedback. We'd like to address some factual inaccuracies in your comments and clarify the misunderstandings.
>
> ## Factual Errors & Misunderstandings
>
> ### Q1. The multi-task Real-SR method seems to be not smart. There are infinite number of degradations. The method randomly samples 100 degradation tasks and obtains 100 fine-tuned models.
>
> Please note that **the 100 degradation tasks are only used for illustration in Figure 1 (Line 115)**. We sampled **4,000** tasks for the main experiments (**Line 198**) and **10,000** tasks for experiments in the supplementary materials (**Line 460**). To efficiently identify the unsatisfactory tasks from such a large set of tasks, we do not train a single-task network for each task, but propose a gradient-based performance indicator as described in **Lines 138-148**. As also noticed by Reviewer yhLt, it is obviously much faster than directly fine-tuning a single-task network.
>
> ### Q2. The performance is not state-of-the-art. (i) In Table 1, the proposed method is worse than RDSR. (ii)In Table 2, RealESRGAN-G has no advantage over RealESRGAN for Group0 and Group1. How to demonstrate the superiority of the proposed method. (iii)In Figure 5, the input images have much noise, not low-resolution. It would be better to provide SR results.
>
> (i) Please note that RDSR is trained with the MSE loss and hence has higher PSNR than GAN-based SR methods, which is normal.
>
> (ii) RealESRGAN-G is not our method but used to analyze the performance of RealESRGAN in different groups (**Lines 272-276**). It can be seen clearly that our TGSR outperforms RealESRGAN.
>
> (iii) Figure 5 shows real-SR results. Please note that real-SR encompasses multiple degradations, including noise. In the supplementary file (**Section 7.4.1 and 7.4.2**), we've provided additional real-SR results, which include degradations with slight noise.
>
> ### Q3. The novelty of this paper should be highlighted. This paper is similar to ClassSR. What if directly compare the PSRN (like ClassSR) with a threshold?
>
> We'd like to highligt the novelty of our method by clarifying two basic concepts.
> #### 1. **Efficient-SR vs. Real-SR**
> (1) We are very familiar with ClassSR, which solves the efficient SR problem under the classical SR setting and only has one degradation task, i.e., bicubic down-sampling. Hence, one GT image corresponds to one LR image. ClassSR divides the LR images into simple, middle, and hard texture groups by absolute PSNR value.
>
> (2) Real-SR (e.g., BSRGAN and Real-ESRGAN) introduces a complex degradation model to synthesize LR images, which include a series of degradation types, such as blur, noise, resize, and JPEG. Hence, one GT image corresponds to multiple LR images. It is clear that hard textures may contain slight and strong degradations. Therefore, there is no linear correlation between absolute PSNR values and unsatisfactory degradation tasks.
>
> #### 2. **Absolute PSNR vs. Relative PSNR**
>
> (1) The objective of Real-SR is to achieve improved performance across all degradation tasks. In Section 3, we observe a range of absolute PSNR values, from 19dB to 25dB, on the validation set. It is worth noting that unsatisfactory degradation tasks can encompass both slight (e.g., 24dB) and strong (e.g., 21dB) degradations. Therefore,**relying solely on absolute PSNR values is inadequate for identifying unsatisfactory tasks**.
>
> (2) To address this issue, we utilize the performance difference, specifically the PSNR distance between single-task networks and the multi-task Real-SR network, as a means to identify unsatisfactory tasks. This approach is rooted in the common understanding within the field of multi-task learning [1,2,3] and is exemplified in notable works such as the CVPR 2018 best paper Taskonomy [2]. By considering the performance difference, we can more effectively identify tasks that fall short of expectations and require further attention.
>
> ### Q4. Low noise and strong noise are the same degradations, which should be treated as the same task.
>
> (1) The task grouping approach (e.g., the CVPR-2018 best paper: Taskonomy [2]) widely used in multi-task learning [1,2,3] focuses on finding the tasks that should be trained together based on the performance differences (e.g., PSNR distance) between the multi-task network and single-task networks.
>
> (2) As mentioned in [1], the term “task similarity” can easily be misunderstood to imply a strong attribute relationship between tasks. In fact, the task similarity in our paper represents the affinity relationship between the single-task network and the multi-task (real-SR) network, rather than the similarity between degradation types. We will further clarify this in the final version.
>
> (3) The consensus in the field recognizes that improved performance, such as a higher PSNR, contributes to superior visual outcomes. Therefore, if performance enhancements can be attained, grouping weak blur and noise together is deemed acceptable. Our experimental findings further validate the achievement of state-of-the-art results.
>
> ### Q5. Each degradation of RealESRGAN is a composite function which different degradations. Could you investigate the effect of such a case?
>
> In our response to Q4, we've clarified that our task grouping method is not based on degradation types but performance. In addition, it is difficult to analyze the composite function in a large degradation space, as pointed out in [4].
>
> ## Other Comments
>
> ### Q6. Is the real-SR model fine-tuned on a pre-trained Real-ESRGAN?
>
> In our experiment, the real-SR model is fine-tuned on a pre-trained Real-ESRGAN. We will make this clear in the final version.
>
> [1] Efficiently Identifying Task Groupings for Multi-Task Learning
>
> [2] Taskonomy: Disentangling task transfer learning
>
> [3] Which Tasks Should Be Learned Together in Multi-task Learning?
>
> [4] Crafting Training Degradation Distribution for the Accuracy-Generalization Trade-off in Real-World Super-Resolution

---

> ### Author Response · Authors · 2023-08-20
>
> Dear Reviewer tGSk,
>
> As the discussion period is closing soon, we would really appreciate if you would let us know whether your concerns have been resolved. We would be happy to discuss with you if you have further questions. Thank you very much for your time!
>
> Regards,
>
> Authors

---

### Official Review · Reviewer_LkK8 · 2023-07-11

**Soundness:** 2 fair
**Presentation:** 3 good
**Contribution:** 2 fair
**Rating:** 4
**Confidence:** 5

**Summary:**

This paper casts real-world image SR as a multi-task learning task for complex image degradations. By comparing the performance gap between the real-SR model and the fine-tuned model for a specific degradation, the satisfactory task and unsatisfactory task are divided. And thus it proposes a task grouping method to address real-world SR.

**Strengths:**

+ The paper is well-written and easy to understand.
+ It is interesting to analyze the real-world image SR from a multi-task learning view.
+ The paper has considered many degradations for evaluation.

**Weaknesses:**

- The main concern is about the motivation. As claimed in Line 36-37, real SR faces a challenge of task competition or task conflict. But,
1) it is still blind to explicitly represent or accurately model specific real degradations for the task of real SR. What is the meaning of "task" for real SR? What is the evidence of the "task conflict"?
2) Although authors use the performance gap to define satisfactory task and unsatisfactory task, it seems to have a prerequisite that the results of pre-trained real-SR netwrok and single-task networks on each degradation task are reliable, especially based on their PSNR distance (Line 119).  This is rather unconvincing. It is hard for me to believe this demonstration for the motivation in Sec.3 and Fig.1.
3) Line 126: "Our analysis in Sec. 3 shows that when a real-SR network is tasked with many degradation tasks, they may compete for model capacity or interfere with each other, resulting in a significant decline in performance for certain tasks."  I could not understand what are the evidences for the claimed completition. Besides, the performance decline may be not cetainly resulted from the claimed completition and be influenced by other factors.

Overall, the analyses on the motivation are rather unconvincing and it is hard for me to believe the rationality of the work.

- Despite the analyses in Sec.3, it does not explain what cases the satisfactory task or unsatisfactory task actually indicate (it is too simple based on a PSNR threshold) and why they have these differences. I think this explain is important to well understand the real SR task and the motivation of taking it as a multi-task learning. Besides, how to undestand the derived degradation task groups in Fig.4?

- Method (Sec.4). It could not ensure the reliability of using pre-trained real-sr network and single-task networks.

- Evaluation:
1. Line 204: "we evaluate the model on Set14...." But, Set14 has very limited number of images. Is this concincing?
2. All the evaluations are based on RealESRGAN and there is no any evaluation on real data for real degration, not generated degradation by gan-based model. This is a very important demonstration extension for the work.

-

**Questions:**

I have listed and explained my concerns on this paper in "Weakness". I expect the responses to those issues.

**Limitations:**

Authors just claim one limitation of the sampling from the degradation space in "Conclusion". This is actually an important issue for this work. Thus the more demonstration on real data, not from realESRGAN would be more convincing.

---

> ### Author Rebuttal · Authors · 2023-08-10
>
>
> Dear Reviewer LkK8,
>
> Thank you for your feedback. We’d like to provide some contextual information to help you better understand our method and address your concerns.
>
> ## Task Grouping in Multi-task Learning
>
> (1) The task grouping approach (e.g., the CVPR2018 best paper: Taskonomy [2]) widely used in multi-task learning [1,2,3] focuses on finding the tasks that should be trained together based on the performance differences (e.g., PSNR distance) between the multi-task network and single-task networks.
>
> (2) As mentioned in [1], the term “task similarity” can easily be misunderstood to imply a strong attribute relationship between tasks. In fact, the task similarity in our paper represents the affinity relationship between the single-task network and the multi-task (real-SR) network, rather than the similarity between degradation cases. We will further clarify this in the final version.
>
> ## Concerns
>
> ### Q1. It is still blind to explicitly represent or accurately model specific real degradations for the task of real SR. What is the meaning of "task" for real SR? What is the evidence of the "task conflict"?
>
> We've provided the definition of "task" in **Lines 93-110**. It is important to note that our task grouping method is not based on the similarity of degradation types but performance. The concept of "task conflict/competition" is explained in **Lines 111-124**.
>
> ### Q2. Despite the analyses in Sec.3, it does not explain what cases the satisfactory task or unsatisfactory task actually indicate (it is too simple based on a PSNR threshold) and why they have these differences. Besides, how to understand the derived degradation task groups in Fig.4?
>
> (1) Basically, the unsatisfactory tasks are the degradation cases not well solved by the jointly-trained real-SR network. Please refer to **Lines 120-124** for the explanation of satisfactory and unsatisfactory tasks.
>
> (2) From Table 1, we can see that the performance of unsatisfactory tasks include both simple degradation cases (25dB in Group 4) and difficult degradation cases (21dB in Group 1), while the performance of satisfactory tasks (Group 0) is around 24dB. This observation indicates that the real-SR network tends to learn the degradations of medium difficulty.
>
>
> ### Q3. Although authors use the performance gap to define satisfactory task and unsatisfactory task, it seems to have a prerequisite that the results of pre-trained real-SR netwrok and single-task networks on each degradation task are reliable, especially based on their PSNR distance (Line 119). This is rather unconvincing. It is hard for me to believe this demonstration for the motivation in Sec.3 and Fig.1.
>
> As mentioned in **Line 210**, the pre-trained real-SR model we used is based on RealESRGAN, a highly influential and well-recognized work in the SR field with **over 22k stars on GitHub**. Using a single-task network as an empirical performance upper bound is a common practice, as demonstrated by works such as AdaFM[6], CResMD[7], and the accuracy-generalization trade-off[4].
>
> ### Q4. Method (Sec.4). It could not ensure the reliability of using pre-trained real-sr network and single-task networks.
>
> Please see our response to Q3.
>
> ### Q5. I could not understand what are the evidences for the claimed completition. Besides, the performance decline may be not cetainly resulted from the claimed completition and be influenced by other factors.
>
> As described in **Lines 114-124**, the only variable we changed is the degradation task number. Therefore, the performance drop is only affected by the task number. This observation is consistent with multi-task learning, where this phenomenon is referred to as task competition, as also stated in **Lines 111-114**.
>
> ### Q6. Evaluation: Line 204: "we evaluate the model on Set14...." But, Set14 has very limited number of images. Is this concincing? All the evaluations are based on RealESRGAN and there is no any evaluation on real data for real degration, not generated degradation by gan-based model. This is a very important demonstration extension for the work.
>
> As stated in **Line 202**, we evaluate the model on Set14 for **each unsatisfactory degradation task**. Our main experiment employs 4000 Set14 test sets, each corresponding to a distinct degradation task.
>
> Due to space limitations, we have included the results on real data in the supplementary materials, as detailed in **Section 7.4.3**.
>
> Thank you for the suggestion. Following the approach used by SwinIR and BSRGAN, we generate a new dataset DIV2K\_random by randomly adding degradations sampled from the degradation model of RealESRGAN to the DIV2K\_val dataset. Additionally, we conduct a comparison on the real benchmark AIM2019-val, the test set used for the real SR track in the AIM 2019 Challenge. On both datasets, our TGSR outperforms state-of-the-art methods consistently and significantly. We will include the results in final version.
>
>
>  |          |        | ESRGAN|BSRGAN|RealESRGAN|SwinIR|DASR|TGSR|
>  |  ----  | ----  |  ----  | ----  |  ----  | ----  |  ----  | ----  |
>  | DIV2K\_random| PSNR |20.63 |23.76|23.54|23.13|23.52|**23.84**|
>  | DIV2K\_random| LPIPS |0.6345 | 0.4622|0.4423|0.4432|0.4832|**0.4368**|
>  | AIM2019-val | PSNR | 23.16 | 24.20 | 23.89 | 23.89 | 23.76 | **24.27**  |
>  | AIM2019-val | LPIPS | 0.5500 | 0.4000 | 0.3960 | **0.3870** | 0.4210 | **0.3899** |
>
> [1] Efficiently Identifying Task Groupings for Multi-Task Learning
>
> [2] Taskonomy: Disentangling task transfer learning
>
> [3] Which Tasks Should Be Learned Together in Multi-task Learning?
>
> [4] Crafting Training Degradation Distribution for the Accuracy-Generalization Trade-off in Real-World Super-Resolution
>
> [5] Unsupervised degradation representation learning for blind super-resolution
>
> [6] Modulating image restoration with continual levels via adaptive feature modification layers
>
> [7] Interactive multi-dimension modulation with dynamic controllable residual learning for image restoration

---

> > ### Comment · Reviewer_LkK8 · 2023-08-21
> >
> > Thanks so much for the authors' responses and so sorry for my late comment.
> > I have carefully read the rebuttal, but it's a pity that my concerns are not well addressed.
> >
> > For example, about "task" and "task conflict", I carefully read the whole paper when reviewing. I am concerned about their unconvincing and intuitive descriptions without evidence and thus I have the question (Q1). But the authors just provide the details in the paper: Lines 93-110, 111-124.
> > Besides, for Q6, for the results of "real data", I have checked the main paper and the supplementary. It still does not provide the result of real data, e.g., RealSR (ICCV2019), not data from RealESRGAN or AIM2019-val.
> >
> > Overall, I still keep my rating.

---

> > > ### Author Response · Authors · 2023-08-21
> > > **Please take a moment to review our additional response**
> > >
> > > Dear Reviewer LkK8,
> > >
> > > Thank you for your response and for highlighting the specific areas that you find unclear. We appreciate your valuable feedback and would like to provide further clarification to address your questions. We sincerely hope that you will take our additional response into consideration and reconsider your evaluation.
> > >
> > > Q1. Unconvincing and intuitive descriptions of "task" and "task conflict" for real SR.
> > >
> > > >In simple terms, an SR task involves restoring a low-resolution image that has undergone a certain degradation. Real-SR assumes that a clear image can undergo various degradations sampled from a large degradation space. For instance, if there are 1000 different possible degradations in the degradation space, a real-SR network is trained to simultaneously address these 1000 degradation tasks, making it a form of multi-task learning. The objective is to enable the network to handle a wide range of degradations. We'd like to emphasize that our formulation of real-SR as a multi-task learning problem (Sec. 3) is rigorous.
> > >
> > > >Task conflict/competition is a fundamental challenge in multi-task learning. It stems from limited model capacity and shared resources. When a model is trained to simultaneously solve multiple tasks, the model must allocate its finite resources, like parameters and computational capacity, to handle each task effectively. However, this allocation creates a trade-off where optimizing one task may come at the expense of others. Figure 1 provides clear evidence that the trained real-SR (multi-task) network favors some degradation tasks over the others, i.e., it is capable of producing satisfactory results for only half of the degradation tasks, while falling short in the remaining half of the degradation tasks.
> > >
> > > >We hope that our explanation has adequately addressed your concerns. Please do let us know if you have any further questions.
> > >
> > > Q2: For Q6, for the results of "real data", I have checked the main paper and the supplementary. It still does not provide the result of real data.
> > >
> > >  >Please note that we've provided results on real-world images in **Section 7.4.3 (page 19) of the Supplementary Material**. These real-world images come from the test set Realworld38, which contains many **real scenes**, such as **old photo scene** (rows 1&2 on page19), **building scene** (row 3 on page19), **greyscale scene** (row 4 on page19), and **web scene** (i.e., images downloaded from the internet) (row 5 on page19). Realworld38 was previously employed in widely recognized studies like SwinIR [1] and DASR [2], and we have merely followed these popular works. For example, the real image OST_009 (which is used in Figure 5 of SwinIR) can be found in the third row of Figure 15 in Section 7.4.3 of our supplementary material (page 19).
> > >
> > > >We apologize for the oversight in not mentioning Realworld38. We will include the information in the final version.
> > >
> > > [1] Liang J, Cao J, Sun G, et al. Swinir: Image restoration using swin transformer[C]//Proceedings of the IEEE/CVF international conference on computer vision. 2021: 1833-1844.
> > >
> > > [2] Liang J, Zeng H, Zhang L. Efficient and degradation-adaptive network for real-world image super-resolution[C]//European Conference on Computer Vision. Cham: Springer Nature Switzerland, 2022: 574-591.

---

> ### Author Response · Authors · 2023-08-20
>
> Dear Reviewer LkK8,
>
> As the discussion period is closing soon, we would really appreciate if you would let us know whether your concerns have been resolved. We would be happy to discuss with you if you have further questions. Thank you very much for your time!
>
> Regards,
>
> Authors

---

### Decision · Program_Chairs · 2023-09-21

**Decision:**

Accept (poster)

**Comment:**

The paper received mixed reviews from five reviewers and the authors provided responses to the expressed concerns.

Reviewer LkK8 is Borderline Reject and keeps his rating after rebuttal considering that his concerns are not well addressed.

Reviewer tGSk is Reject and keeps his rating after rebuttal.

Reviewer 4Sx7 upgrades to Borderline Accept after rebuttal.

Reviewer yhLt upgrades to Strong Accept after rebuttal.

Reviewer o9KK keeps his Borderline Accept after rebuttal.

After carefully reading the manuscript, the reviews, the authors' responses and the discussions, it is clear that the authors did a good job with their rebuttal and convinced two reviewers to improve their ratings. The negative ("Reject") is from Reviewer tGSk but, unfortunately, did not provide a final comment to the authors' response, while the other reviewers are borderline accept (4Sx7 and o9KK), borderline reject (LkL8) and strong accept (yhLT).

The ACs agree with the majority of the reviewers that the work has merits and is of interest for the community and considers that the authors should further refine their manuscript for camera ready by including (part of) contents/info provided in their responses and by benefiting from the reviewers' feedback.